# CLCa mediates a novel cross-talk between Wnt secretion and actin organization

Mahak Tiwari[1,2], Mihir Dingankar[3], Jyoti Das[1,2], Sreelekshmi S R[1], Apurv Solanki[1], Deepa Subramanyam[1]

Mammalian clathrin light chains (CLCa, CLCb) are critical players in clathrin-mediated endocytosis. However, their physiological role in contributing to specific cellular processes and early development remains elusive. To elucidate their individual functions, we generated CLC knockout mESCs. Loss of CLCa resulted in down-regulation of Wnt pathway genes along with altered secretion of Wnt3a because of impaired trafficking of its secretion mediator, WLS. Reduced Wnt signaling led to lower levels of Hip1R causing a reorganization of the actin cytoskeleton. CLCa knockout cells displayed actin patches enriched for Arp3 and cortactin, with activation of the Wnt pathway resulting in disassembly of these patches. Furthermore, we uncovered a bidirectional cross-talk between Wnt signaling and actin organization, with actin disruption resulting in lower Wnt signaling. Our data reveal a previously undiscovered role of CLCa in mediating molecular communication between actin organization and Wnt signaling.

## Introduction

The trafficking of molecules between membrane-bound structures often proceeds with the assembly of specific coat proteins. One such process is clathrin-mediated endocytosis, involving the coat protein, clathrin. Clathrin forms a three-legged structure (triskelion), which is the basic building block in the generation of the clathrin coat. The triskelion in turn is composed of three molecules each of the clathrin heavy chain (CHC) and of the clathrin light chain (CLC). Although the function of CHC is undisputed in the context of endocytosis (Payne et al, 1988; Royle, 2006; Brodsky, 2012; Briant et al, 2020), the role of the light chain is still largely elusive. Existing evidence points toward a role of CLCs in the regulation of assembly and disassembly of clathrin coats (Wilbur et al, 2010; Brodsky, 2012; Dannhauser et al, 2015), internalization of specific G protein–coupled receptors (Maib et al, 2018), B-cell maturation in the germinal center, antibody isotype switching (Wu et al, 2016), regulation of epithelial lumen formation (Chen et al, 2024), actin

organization (Chen & Brodsky, 2005; Poupon et al, 2008; Wilbur et al, 2008; Majeed et al, 2014; Mukenhirn et al, 2021), cell spreading and migration (Majeed et al, 2014; Tsygankova & Keen, 2019), regulation of receptor trafficking (Poupon et al, 2008; Ferreira et al, 2012; Majeed et al, 2014; Wu et al, 2016), and synaptic vesicle recycling (Redlingshöfer et al, 2020).

Higher vertebrates have two light chain genes, *Clta* and *Cltb*, encoding CLCa and CLCb, respectively (Das et al, 2021). Both CLCs have 60% sequence similarity and have characteristic tissue-specific expression. Knockouts of individual light chains in mice showed altered numbers of synaptic vesicles in distinct regions of the brain, accompanied by electrophysiological defects in a light chain–specific manner (Redlingshöfer et al, 2020), highlighting distinct roles for CLCa and CLCb.

Early development is a finely orchestrated process, which can be studied extensively using embryonic stem cells. Self-renewal and pluripotency of mESCs are dependent on a wide variety of cellular and biological processes such as cell signaling pathways (Boyer et al, 2005; Tiwari et al, 2021; Varzideh et al, 2023), endocytic mechanisms (Li et al, 2010; Subramanyam et al, 2011; Qin et al, 2014; Mote et al, 2017; Dambournet et al, 2018), and actin organization (Xia et al, 2019). Previous data from our laboratory have reported a role of the CHC in the maintenance of pluripotency of mESCs (Narayana et al, 2019) and its impact on the reorganization of actin cytoskeleton because of alterations in the pluripotent status of these cells (Mote et al, 2020). However, the individual roles and physiological functions of light chains in pluripotent cells such as mESCs remain unknown. To address these questions, and identify signaling pathways using specific CLCs, we generated individual and double knockout mESC lines for each of the CLCs (*Clta* and *Cltb*) using the CRISPR-Cas9 genome editing approach. Our results revealed that loss of CLCa caused altered trafficking of the Wnt ligand secretion mediator, Wls, through the Golgi, resulting in reduced Wnt secretion and the downstream signaling output of the pathway. Furthermore, cells lacking CLCa also showed altered actin organization with the appearance of actin patches at the basal surface of cells in a Hip1R-dependent manner. We further uncovered a novel cross-talk between Wnt signaling and the actin cytoskeleton via CLCa, through which aberrant actin organization in CLCa knockout cells could be restored by activation of the Wnt

[1]National Centre for Cell Science, SP Pune University Campus, Pune, India   [2]SP Pune University, Pune, India   [3]Indian Institute of Science Education and Research (IISER) Pune, Pune, India

Correspondence: deepa@nccs.res.in

signaling pathway. In addition, disruption of the actin cytoskeleton also resulted in reduced Wnt signaling. We herein describe a regulatory loop wherein removal of CLCa results in altered Wnt secretion and a lowering of Hip1R levels, ultimately resulting in a disorganized actin cytoskeleton. Our results uncover novel and distinct roles of the CLCs in the context of embryonic stem cells, indicative of their importance during early development.

# Results

### Generation and characterization of CLC knockout mESCs

To investigate the distinct physiological roles of CLCa and CLCb in mESCs, we generated individual light chain KO mESC lines ($Clta^{-/-}$ and $Cltb^{-/-}$) and double knockouts for both the light chains ($Clta\_b^{-/-}$) using the CRISPR-Cas9 genome editing system. We generated two independent clones for each KO mESC line. Western blotting for CLCa and CLCb proteins in the KO clones demonstrated the specific loss of the indicated protein (Fig 1A). All the CLC KO mESCs were able to form compact colonies similar to WT mESCs (Fig S1A). We further characterized the CLC knockout mESC lines for the expression of pluripotency and differentiation markers. CLC KO mESC lines did not show any significant change in the expression levels of pluripotency markers *Oct4*, *Sox2*, *Nanog*, and *Klf4* (Fig S1B). However, there were significant changes in the levels of differentiation markers in specific KO mESC lines. $Cltb^{-/-}$ mESCs showed significant up-regulation of endoderm markers *Gata4* and *Gata6* (Fig S1C). We further assessed the differentiation potential of CLC KO mESCs through embryoid body (EB) formation. $Cltb^{-/-}$ mESC–derived EBs showed altered differentiation compared with WT mESC–derived EBs, with features of cystic embryoid bodies with visceral yolk sac–like structures (Yasuda et al, 2009) (Fig S1D). In addition to yolk sac–like structures, $Cltb^{-/-}$ EBs also showed enhanced beating EB foci compared with WT EBs (Fig S1E), with elevated expression of cardiac genes such as *Mlc-2v*, *α-MHC*, and *cTnI* (Fig S1F). $Clta^{-/-}$ EBs retained the elevated expression of pluripotency markers *Oct4*, *Sox2*, and *Nanog* compared with WT EBs (Fig S1G). $Cltb^{-/-}$ EBs showed up-regulated expression of late-stage endoderm markers *Gata4* and *Afp* compared with WT EBs, suggesting a preference toward the endodermal lineage (Fig S1H), similar to what was observed in the undifferentiated mESCs (Fig S1C). To ensure that these observations were not restricted to a single clone, we generated a second clone for each genotype and observed similar results with respect to pluripotency and differentiation marker expression (Fig S2A–D). Directed neuronal differentiation of CLC KO mESCs also showed down-regulated expression of neuronal markers such as *Map2* and *β*-III *tubulin* compared with WT mESC–derived EBs (Fig S1I). In order to assess whether endocytosis of ligands was altered in the KO cells, we used fluorescently labeled transferrin to determine uptake efficiency. The uptake of transferrin was unaffected in all the KO mESCs compared with WT mESCs, indicating that endocytosis in general was unaffected upon loss of CLCs (Fig S1J and K). Our results reveal that specific CLCs may regulate early developmental decisions to specify cell fate. We further speculate that this may arise because of differential trafficking of specific receptors.

### $Clta^{-/-}$ and $Clta\_b^{-/-}$ mESCs show significant down-regulation of Wnt/$β$-catenin pathway genes

To determine whether the CLC KO mESCs displayed an altered gene expression profile, we performed unbiased RNA sequencing and analysis. All the CLC KO mESCs showed differentially expressed genes compared with WT mESCs (Fig 1B, Table 1). Gene ontology (GO) analysis showed enrichment of biological processes involved in tissue and organ development in all the CLC KO mESCs (Fig S3A–C). Similar to what we observed through qRT–PCR analysis (Fig S1C), RNA sequencing data also showed significant up-regulation of various endoderm markers such as *Gata4*, *Gata6*, *Sox7*, and *Sox17* in $Cltb^{-/-}$ mESCs, compared with WT mESCs (Fig S3D). KEGG pathway analysis performed on the differentially expressed genes revealed enrichment for the Wnt signaling pathway in $Clta^{-/-}$ and $Cltb^{-/-}$ mESCs (Fig S3A and B). Examination of the expression of specific components of the Wnt signaling pathway and its targets revealed significant down-regulation of genes such as *Axin2*, *Lef1*, *Wls*, *Dkk1*, in $Clta^{-/-}$ and $Clta\_b^{-/-}$ mESCs (Fig 1C). We further validated the expression of these genes using quantitative RT–PCR in both clones of $Clta^{-/-}$, $Cltb^{-/-}$, and $Clta\_b^{-/-}$ mESCs, which confirmed the reduced expression of *Axin2*, *Lef1*, *Wls*, *Wisp1*, and *Dkk1* in $Clta^{-/-}$ and $Clta\_b^{-/-}$ mESCs (Figs 1D and S4A).

The activity and activation of downstream targets in the activated Wnt pathway are dependent on the stability of $β$-catenin and its translocation from the cytoplasm to the nucleus. Under basal conditions, a "destruction complex" comprising of two Ser/Thr kinases, glycogen synthase kinase-3 (GSK-3) and casein kinase-I, and two scaffold proteins, Axin and APC, is involved in the degradation of $β$-catenin. Binding of Wnt ligands to the Lrp5/6 and Fzd receptors leads to inactivation of this destruction complex, allowing $β$-catenin to accumulate, translocate to the nucleus, interact with TCF/LEF, and activate transcription of target genes. We therefore looked at $β$-catenin transcriptional activity using the TOPFlash luciferase–based reporter assay in KO mESC lines. In KO mESCs, $β$-catenin transcriptional activity was significantly reduced compared with WT mESCs (Fig 1E). Reduced transcriptional activity of $β$-catenin suggests that there may be insufficient $β$-catenin for the activation of the signaling pathway. Interestingly, total protein levels of $β$-catenin remained unchanged in the CLC KO mESCs (Fig S6A and B). To assess whether the signaling pathway was responsive downstream of ligand binding to the receptor, we treated KO mESCs with the GSK3$β$ inhibitor, CHIR99021. GSK3$β$ is a kinase and is a part of the destruction complex. It phosphorylates $β$-catenin in the cytoplasm targeting it for ubiquitin-mediated degradation. Inhibition of GSK3$β$ using CHIR99021 resulted in the prevention of $β$-catenin degradation and rescued $β$-catenin transcriptional activity, measured by the TOPFlash reporter assay (Figs 1E and S4B). In addition to rescuing $β$-catenin transcriptional activity, the addition of CHIR99021 to WT and CLC KO mESCs resulted in the increased expression of Wnt target genes such as *Axin2* and *Sp5* in $Clta^{-/-}$ mESCs compared with $Clta^{-/-}$ mESCs treated with DMSO (Fig S4D and E). However, we did not see a rescue in the transcript levels of *Lef1*; instead, we observe a greater down-regulation upon treatment with CHIR99021 (Fig S4F). The addition of exogenous Wnt ligand also resulted in rescued $β$-catenin transcriptional activity (Figs 1F and S4C). Furthermore, we added

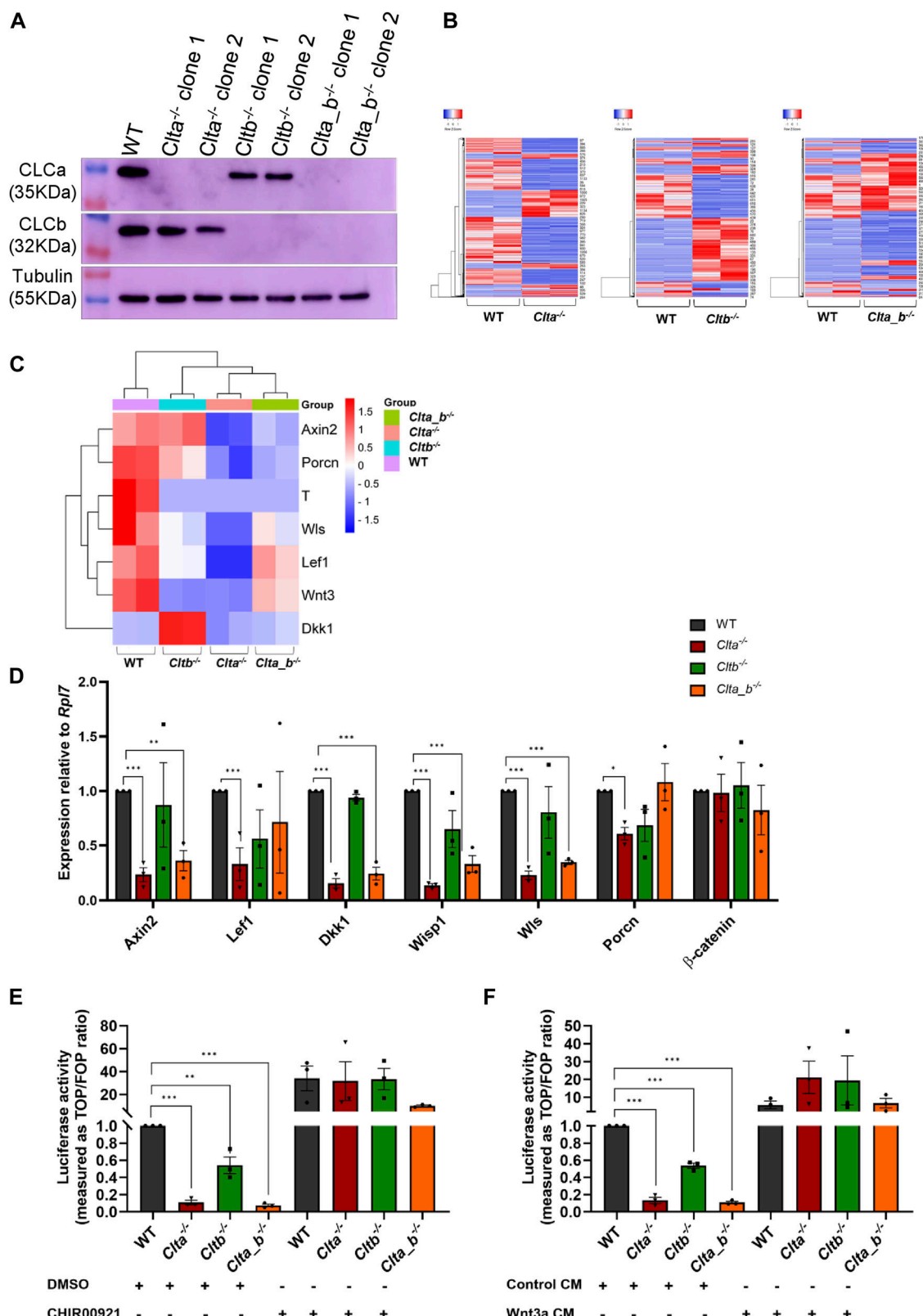

**Figure 1. Wnt/β-catenin signaling pathway is affected in *Clta⁻/⁻* and *Clta_b⁻/⁻* mESCs.**
**(A)** Western blots showing the expression of CLCa, CLCb, and tubulin in WT, *Clta⁻/⁻*, *Cltb⁻/⁻*, and *Clta_b⁻/⁻* mESCs. **(B)** Heatmap indicating differentially expressed genes in *Clta⁻/⁻*, *Cltb⁻/⁻*, and *Clta_b⁻/⁻* mESCs compared with WT mESCs. Red represents more highly expressed genes, whereas blue indicates genes with a lower expression level. **(C)** Heatmap depicting the expression of Wnt/β-catenin pathway and target genes in *Clta⁻/⁻*, *Cltb⁻/⁻*, and *Clta_b⁻/⁻* mESCs. **(D)** qRT–PCR analysis of Wnt/β-catenin pathway

recombinant Wnt3a to differentiating EBs and performed qRT-PCR to check for the expression of pluripotency and differentiation markers. As mentioned previously, *Clta*$^{-/-}$ mESC–derived EBs retained the expression of pluripotency markers such as Oct4, Sox2, and Nanog compared with WT mESC–derived EBs (Fig S1G). Upon the addition of Wnt3a, the expression of Oct4 and Sox2 was found to be reduced in differentiated *Clta*$^{-/-}$ mESC–derived EBs compared with the control *Clta*$^{-/-}$ mESC–derived EBs (Fig S5A and B). However, we did not see significant changes in the expression of differentiation markers such as *Gata4*, *Gata6*, *T-bra*, and *Nestin* upon the addition of Wnt3a (Fig S5C–F). Thus, although certain aspects of pluripotency were rescued upon the addition of exogenous Wnt3a, not all features were rescued, indicative of the involvement of additional molecules and pathways that may be altered in the absence of CLCa. Also, the rescue of β-catenin transcriptional activity upon exogenous Wnt3a addition indicates the presence of receptors on the membrane and hence activation of the downstream signaling pathway. To assess this, we determined the total amount of Lrp6 in these cells. Interestingly, we observed a significant decrease in total protein levels of Lrp6 in *Clta*$^{-/-}$ mESCs (Fig S6C and D). The rescue of β-catenin transcriptional activity either by inhibition of GSK3β or by the addition of exogenous Wnt ligand further suggested that even though a decrease in the levels of the receptors was observed, this may not be the major driver for the reduced activity of the Wnt pathway in the absence of specific CLCs.

### Secretion of Wnt3a is altered in *Clta*$^{-/-}$ and *Clta_b*$^{-/-}$ mESCs

As mentioned above, β-catenin transcriptional activity could be rescued upon treatment with the GSK3β inhibitor, CHIR99021, or upon the exogenous addition of Wnt3a. Our mRNA sequencing and qRT–PCR analysis also revealed that genes involved in the trafficking and secretion of Wnt were also affected upon loss of CLCa. We therefore speculated whether the secretion of Wnt3a may be affected in *Clta*$^{-/-}$ and *Clta_b*$^{-/-}$ mESCs. All Wnts have signal peptides that direct the newly synthesized protein to the lumen of the ER, where they are palmitoylated (Rios-Esteves & Resh, 2013) with the help of an ER-resident membrane-bound O-acyltransferase, porcupine (Porcn). The lipid-modified Wnts are then transported to the cell surface by the carrier protein, WLS (Wntless) (Yu et al, 2014). This indicates that for the secretion of Wnts from the ER, and their transport to the cell surface, PORCN and WLS are required. *Clta*$^{-/-}$ and *Clta_b*$^{-/-}$ mESCs showed down-regulated mRNA levels for *Wls* (Fig 1C and D). We therefore asked whether Wnt secretion was altered in the absence of CLCs. To study the same, we made knockouts for the light chains in cells engineered to secrete high levels of Wnt3a (L cells overexpressing Wnt3a) (Fig S7A). Using an ELISA-based assay, we checked the secretion of Wnt3a in CLC KO Wnt3a L cells. Wnt3a secretion was

significantly reduced in *Clta*$^{-/-}$ and *Clta_b*$^{-/-}$ Wnt3a L cells at early time points (6 h, 12 h) (Fig 2A) and even at 24 h (Fig S8A), compared with later time points (48 h) (Fig S8B). The total level of Wnt3a was, however, unaltered in these cells (Fig S7B). This indicated that *Clta*$^{-/-}$ and *Clta_b*$^{-/-}$ Wnt3a L cells had a delayed and inefficient secretion of Wnt3a compared with WT Wnt3a L cells. As a control, we also treated WT and CLC KO Wnt3a L cells with ETC-159, a small molecule porcupine inhibitor, resulting in complete inhibition of Wnt secretion (Fig S8C).

Clathrin is involved in the trafficking of molecules from the Golgi to the PM, endosomes, and lysosomes (Abazeed et al, 2005; Polishchuk et al, 2006; Radulescu et al, 2007; Braulke & Bonifacino, 2009). However, the specific involvement of CLCs in this process is poorly studied. A report from Poupon et al (2008) showed that CLCs regulate the trafficking of proteins such as CI-MPR from the Golgi. To look at the role of CLC in the trafficking of WLS from the Golgi, we overexpressed mCherry-tagged WLS in these cells to determine its localization. WLS was localized at the cis-medial Golgi, where it colocalized with CLCa (Fig 2B), suggesting that its trafficking from the Golgi could occur in a CLCa-dependent manner. We then looked for the presence of WLS at the Golgi in CLC KO Wnt3a L cells. WLS was found to accumulate in the Golgi in *Clta*$^{-/-}$ and *Clta_b*$^{-/-}$ Wnt3a L cells compared with WT or *Cltb*$^{-/-}$ Wnt3a L cells (Fig 2C and D). This indicated that the delay in secretion of Wnt3a in *Clta*$^{-/-}$ and *Clta_b*$^{-/-}$ Wnt3a L cells could result from the impaired trafficking of WLS, resulting in a lower amount of Wnt3a reaching the plasma membrane in *Clta*$^{-/-}$ Wnt3a L cells, compared with WT Wnt3a L cells. To address this, we performed TIRF microscopy and quantified the amount of Wnt3a present at the plasma membrane (PM). The amount of Wnt3a at the PM was reduced in *Clta*$^{-/-}$ and *Clta_b*$^{-/-}$, compared with WT Wnt3a L cells (Fig 2E and F). Furthermore, to confirm the involvement of CLCa in Wnt3a secretion, we overex-pressed CLCa in WT and CLC KO Wnt3a L cells and examined the levels of Wnt3a at the plasma membrane by TIRF microscopy. The overexpression of CLCa in *Clta*$^{-/-}$ and *Clta_b*$^{-/-}$ Wnt3a L cells resulted in the increased presence of Wnt3a at the PM, compared with control knockout cells (Fig S8D and E), indicating a specific role of CLCa in the secretion of Wnt3a. Together, our observations in-dicate that CLCa regulates the trafficking of WLS and Wnt3a in the Golgi, thereby regulating Wnt3a secretion and downstream pathway activation. Needless to say, such altered signaling can affect the outcome of developmental decisions.

### Actin organization is altered in *Clta*$^{-/-}$ and *Clta_b*$^{-/-}$ mESCs

It has been previously shown that knocking down both the light chains resulted in impaired trafficking of CI-MPR from the TGN in HeLa cells and also resulted in altered actin organization (Poupon et al, 2008). Along with CLC role in the maintenance of actin

---

and target genes in WT, *Clta*$^{-/-}$, *Cltb*$^{-/-}$, and *Clta_b*$^{-/-}$ mESCs. mRNA expression is normalized to *Rpl7* and represented relative to WT mESCs. **(E)** TOPFlash luciferase reporter assay showing levels of β-catenin–mediated transcriptional activation in WT, *Clta*$^{-/-}$, *Cltb*$^{-/-}$, and *Clta_b*$^{-/-}$ mESCs, treated with DMSO or CHIR99021. **(F)** TOPFlash reporter assay showing levels of β-catenin–mediated transcriptional activation in WT, *Clta*$^{-/-}$, *Cltb*$^{-/-}$, and *Clta_b*$^{-/-}$ mESCs, treated with control CM or Wnt3a CM collected at 72 h from L cells and Wnt3a L cells, respectively. Error bars represent the mean ± SEM for experiments (N = 3). *$P < 0.05$; **$P < 0.01$; ***$P < 0.001$ by $t$ test. Source data are available for this figure.

**Table 1.** Table showing the number of differentially expressed genes in $Clta^{-/-}$, $Cltb^{-/-}$, and $Clta\_b^{-/-}$ mESCs compared with WT mESCs.

| | Up-regulated | Down-regulated | Baseline | Total significant | DEG |
|---|---|---|---|---|---|
| Clta-/- | 334 | 808 | 6,234 | 7,376 | 21,793 |
| Cltb-/- | 383 | 305 | 4,831 | 5,519 | 22,040 |
| Clta_b-/- | 285 | 436 | 5,651 | 6,372 | 22,036 |

organization, both canonical and noncanonical Wnt signaling are also involved in actin organization and assembly (Lai et al, 2009; Hajka et al, 2021). In canonical Wnt signaling, GSK3$\beta$ is shown to be involved in regulating actin filament organization and focal adhesion by regulating Arp2/3 activity in rat fibroblasts (To et al, 2017). Various other reports have also demonstrated a role of CLCs in actin organization in differentiated cells (Chen & Brodsky, 2005; Majeed et al, 2014; Mukenhirn et al, 2021). We speculated that in the absence of CLCs, altered Wnt signaling may affect actin organization in a GSK3$\beta$-dependent manner. Analysis of actin organization revealed that $Clta^{-/-}$ and $Clta\_b^{-/-}$ mESCs had an increased number of actin patches at the basal cell surface compared with WT mESCs (Figs 3A and B and S9A and B). Interestingly, cells containing a greater number of patches displayed a reduction in the actin mesh density (Fig 3C) and increased cell area (Fig 3D). Previous studies have demonstrated that a reduced actin mesh density correlated with a greater degree of pluripotency (Xia et al, 2019). This is in line with our previous observations where even under differentiating conditions, mESCs lacking CLCa retain the expression of pluripotency markers (Fig S1G).

At their N terminus, CLCs have a consensus sequence of 22 amino acids through which they bind to actin via Hip1/Hip1R proteins (Chen & Brodsky, 2005). At sites of endocytosis, CLCs recruit actin via Hip1R, which in turn recruits actin binding proteins such as cortactin and the Arp2/3 complex to form a branched actin network around the clathrin-coated pit (CCP). This helps the CCP to invaginate from the PM. The Arp2/3 complex is also involved in regulating the actin mesh density in mESCs (Xia et al, 2019). As the actin organization was altered in $Clta^{-/-}$ and $Clta\_b^{-/-}$ mESCs, we determined the levels of specific actin and actin binding proteins. $Clta^{-/-}$ and $Clta\_b^{-/-}$ mESCs showed a reduction in levels of Hip1R, Hip1, and actin itself compared with WT mESCs (Fig 3E and F). However, the levels of cortactin and Arp3 remained unchanged in these KO mESC lines. Engqvist-Goldstein et al earlier reported that reduced Hip1R levels resulted in enrichment of cortactin and Arp2/3 in actin patches. In line with this, we observed enrichment of cortactin and Arp3 in the actin patches in $Clta^{-/-}$ and $Clta\_b^{-/-}$ mESCs (Fig 3G and H). This sequestration of Arp2/3 complex components and cortactin into the patches could result in their reduced cytoplasmic availability, thus altering the actin meshwork in these cells. Previous studies indicate that Hip1R interacts with cortactin to inhibit actin assembly, with depletion of Hip1R resulting in over-assembly of actin structures driven by Arp2/3 (Le Clainche et al, 2007). Despite sharing the same consensus sequence at the N terminus, both the CLCs appear to regulate actin organization differentially. In WT mESCs, both CLCa and CLCb bind to Hip1 and Hip1R (Fig S10), but only the absence of CLCa resulted in reduced Hip1R levels (Fig 3E and F) and altered actin organization (Fig 3A). Yeast has a single gene encoding for the CLC, which shares greater homology with CLCa than CLCb. In a similar manner, actin-dependent endocytic events in mESCs may prefer CLCa to CLCb. This may be due to functional differences in the sequences outside of the common Hip1R binding region, which affect Hip1R–actin interactions. This therefore suggests that the formation of actin patches may be driven by a decrease in Hip1R levels because of the specific loss of CLCa.

### Reduction in actin patches in $Clta^{-/-}$ and $Clta\_b^{-/-}$ cells upon the overexpression of Hip1R or CLCa

To determine whether the generation of actin patches was specific to mESCs in the absence of CLCs, or whether similar phenomena were also observed in other cells, we generated KO lines in mouse L cells (fibroblast cells) using the same CRISPR-Cas9 genome editing strategy (Fig S11A). Both $Clta^{-/-}$ and $Clta\_b^{-/-}$ L cells showed the presence of actin patches at the basal cell surface, similar to what was observed in mESCs (Fig S11B). These patches were also enriched for cortactin and Arp3 (Fig S11C and D). To further validate the role of CLCa in actin patch formation via Hip1R, we performed rescue experiments in CLC KO cells. The overexpression of either CLCa or Hip1R reduced the formation of actin patches in $Clta^{-/-}$ and $Clta\_b^{-/-}$ L cells (Figs 4A and B and S11E). The overexpression of Hip1R in $Clta^{-/-}$ and $Clta\_b^{-/-}$ mESCs also reduced the actin patches (Fig 4C and D). Together, these results further validated that the loss of CLCa specifically resulted in reduced levels of Hip1R causing an altered organization of the actin cytoskeleton.

### Cross-talk between Wnt signaling and actin reorganization via CLCa

As mentioned above, $Clta^{-/-}$ and $Clta\_b^{-/-}$ cells showed reduced Wnt3a secretion and Wnt/$\beta$-catenin signaling, along with altered actin structures. We speculated whether a three-way connection may exist between CLCa, Wnt signaling, and actin. $Clta^{-/-}$ and $Clta\_b^{-/-}$ Wnt3a L cells did not show the presence of actin patches at the basal cell surface compared with $Clta^{-/-}$ and $Clta\_b^{-/-}$ L cells, which do not secrete Wnt3a (Fig 5A and B). This suggested that the presence of excess Wnt3a may prevent the reorganization of the actin cytoskeleton even in the absence of CLCa. It has been previously shown that Wnt3a can reorganize the actin cytoskeleton when L cells were treated with Wnt3a CM. Stress fibers in L cells were arranged in multiple directions, whereas in Wnt3a-CM-treated cultures, the actin stress fibers appeared to be aligned along a single direction (Shibamoto et al, 1998). Similar to previous observations, we observed parallelly arranged stress fibers in Wnt3a L cells, compared with L cells (Fig 5A). In addition to stress fiber rearrangement, actin patches were also present in $Clta^{-/-}$ and

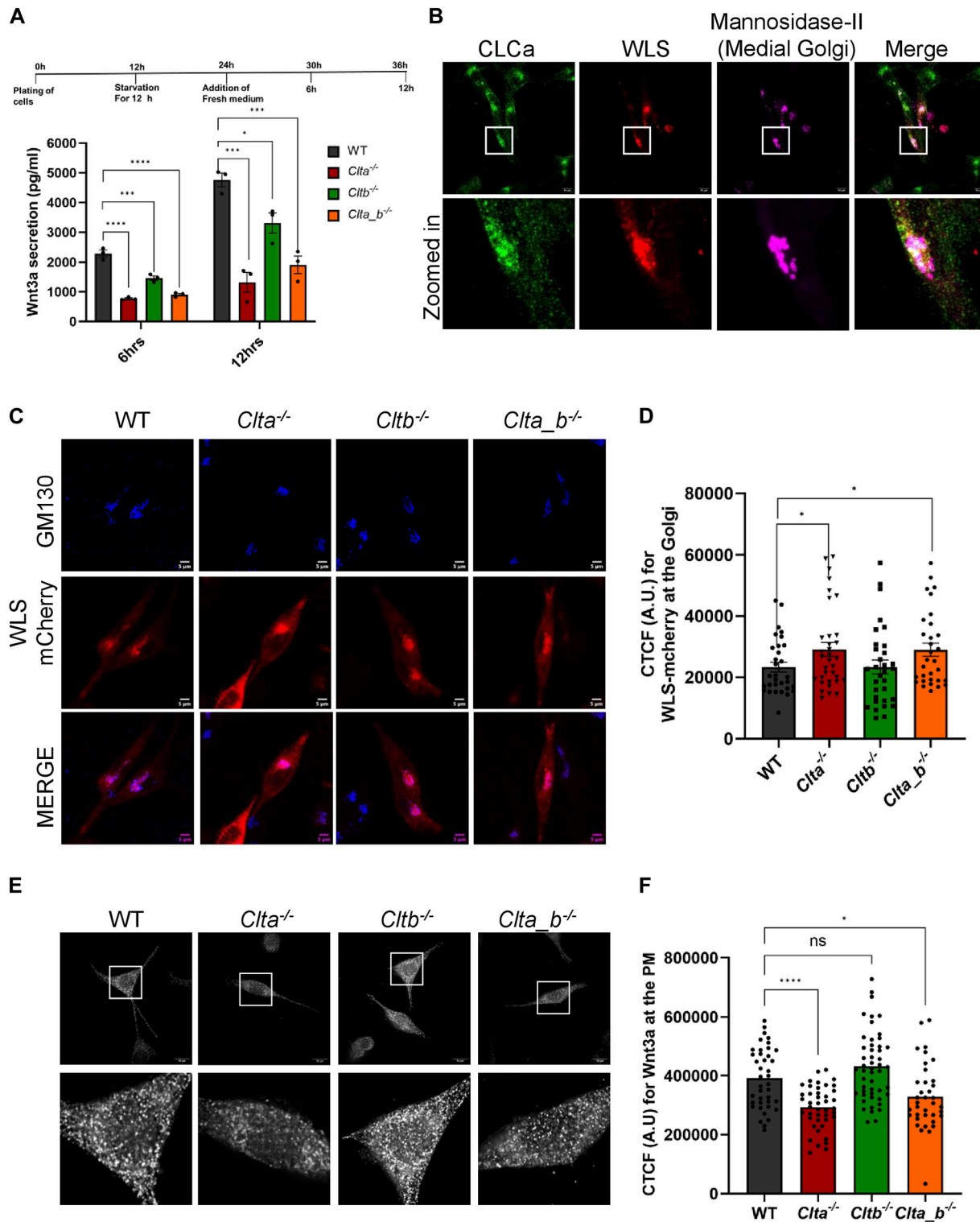

**Figure 2. Wnt3a secretion is reduced in *Clta*<sup>−/−</sup> and *Clta_b*<sup>−/−</sup> cells because of impaired trafficking of its carrier protein WLS.**
**(A)** Quantitation of secreted Wnt3a from WT, $Clta^{-/-}$, $Cltb^{-/-}$, and $Clta\_b^{-/-}$ Wnt3a L cells at indicated time points by ELISA. **(B)** Confocal images showing immunostaining for CLCa in Wnt3a L cells transfected with WLS-mCherry and mannosidase II-GFP plasmids (scale bar—10 $\mu$m). **(C)** Confocal images acquired after immunostaining for GM130 (Golgi marker) in WT, $Clta^{-/-}$, $Cltb^{-/-}$, and $Clta\_b^{-/-}$ Wnt3a L cells, transfected with the WLS-mCherry plasmid (scale bar—5 $\mu$m). **(D)** Bar graph showing corrected total cell fluorescence (CTCF) for WLS within the Golgi (marked by cis-Golgi marker GM130) in WT, $Clta^{-/-}$, $Cltb^{-/-}$, and $Clta\_b^{-/-}$ Wnt3a L cells. **(E)** TIRF microscopy for Wnt3a presence at the plasma membrane in WT, $Clta^{-/-}$, $Cltb^{-/-}$, and $Clta\_b^{-/-}$ Wnt3a L cells (scale bar—10 $\mu$m). **(F)** Bar graph showing CTCF values for Wnt3a at the PM in WT, $Clta^{-/-}$, $Cltb^{-/-}$, and $Clta\_b^{-/-}$ Wnt3a L cells. Error bars represent the mean ± SEM for experiments in triplicates (N = 3). *$P < 0.05$; **$P < 0.01$; ***$P < 0.001$ by $t$ test and one-way ANOVA test for Wnt3a secretion. Source data are available for this figure.

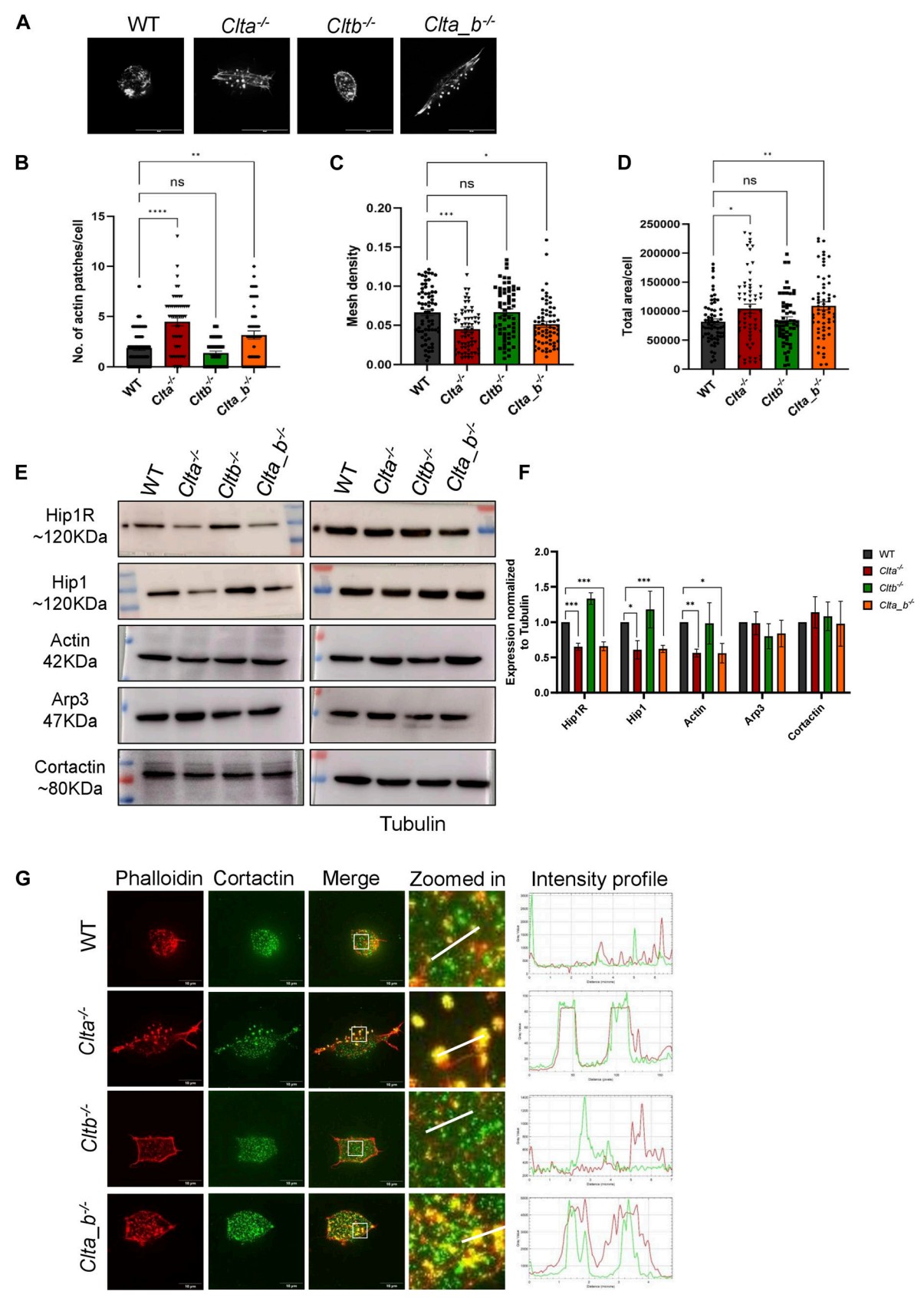

**Figure 3.**
(Continued)

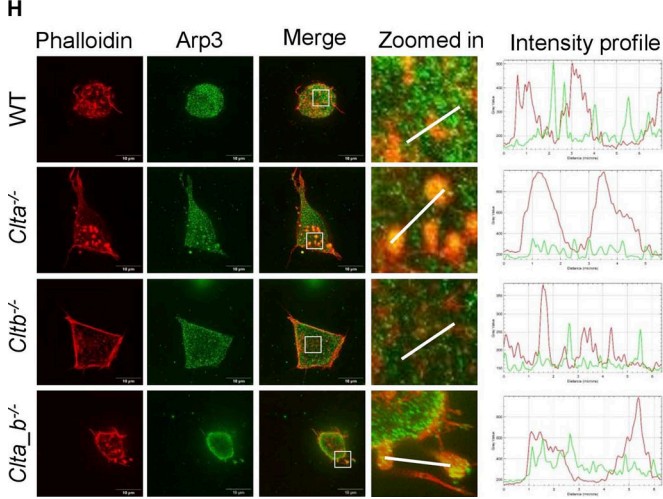

**Figure 3. Actin organization is affected in mESCs lacking CLCa.**
**(A)** Super-resolution spinning disk confocal microscopic images
for visualization of F-actin stained with phalloidin in WT, *Clta⁻/⁻*, *Cltb⁻/⁻*, and
*Clta_b⁻/⁻* mESCs (scale bar—20 µm). **(B, C, D)** Bar graphs representing quantitative
analysis of various actin features: (B) actin patches; (C) mesh density; and (D)
total area per cell for WT, *Clta⁻/⁻*, *Cltb⁻/⁻*, and *Clta_b⁻/⁻* mESCs. **(E)** Western blots
showing the expression of Hip1R, Hip1, actin, Arp3, cortactin, and tubulin in WT,
*Clta⁻/⁻*, *Cltb⁻/⁻*, and *Clta_b⁻/⁻* mESCs. **(F)** Bar graph showing quantitation of total
protein levels of Hip1R, Hip1, actin, Arp3, and cortactin normalized to tubulin for
WT, *Clta⁻/⁻*, *Cltb⁻/⁻*, and *Clta_b⁻/⁻* mESCs. **(G)** Super-resolution spinning disk
confocal microscopic images for visualization of cortactin and F-actin in WT,
*Clta⁻/⁻*, *Cltb⁻/⁻*, and *Clta_b⁻/⁻* mESCs, along with intensity profile graphs for
showing colocalization of cortactin and actin patches stained with phalloidin
(scale bar—10 µm). **(H)** Super-resolution spinning disk confocal microscopic
images for visualization of Arp3 and F-actin in WT, *Clta⁻/⁻*, *Cltb⁻/⁻*, and *Clta_b⁻/⁻*
mESCs, along with intensity profile graphs for showing colocalization of Arp3 and
actin patches stained with phalloidin (scale bar—10 µm). Error bars represent
the mean ± SEM for experiments in triplicates (N = 3). *$P < 0.05$; **$P < 0.01$; ***$P <
0.001$ by *t* test.
Source data are available for this figure.

*Clta_b⁻/⁻* L cells in comparison with *Clta⁻/⁻*and *Clta_b⁻/⁻* Wnt3a L
cells (Fig 5A and B). In line with this, treatment of *Clta⁻/⁻* and
*Clta_b⁻/⁻* mESCs with the GSK3β inhibitor CHIR99021 reduced the
number of actin patches (Fig 5C and D) and restored mesh
density (Fig 5E). However, the addition of recombinant Wnt3a to
mESCs did not promote their survival (data not shown). Treat-
ment of *Clta⁻/⁻* and *Clta_b⁻/⁻* L cells either with CHIR99021 or with
recombinant Wnt3a reduced actin patches in *Clta⁻/⁻* L cells but
not in *Clta_b⁻/⁻* L cells (Figs S12A and B and S14D). This indicated
that additional players beyond just Wnt regulators may be in-
volved in actin organization in the absence of both light chains,
although this is at odds with what we see in Wnt3aL cells, where
patches do not appear even in the absence of both light chains.
In order to determine whether this was exclusively due to Wnt3a,
we blocked Wnt3a secretion using the porcupine inhibitor ETC-
159. Treatment of WT and CLC KO Wnt3a L cells with ETC-159
resulted in the formation of actin patches in both *Clta⁻/⁻* and
*Clta_b⁻/⁻* Wnt3a L cells, with a greater number of *Clta⁻/⁻* cells
presenting patches (Fig S13A). Treatment of these cells for
7 d with ETC-159 further increased the number of cells with
patches (Fig S13B). Washout of the inhibitor for 3 d after 7 d of

treatment caused reduction in the numbers of actin patches in
*Clta⁻/⁻* and *Clta_b⁻/⁻* Wnt3a L cells (Fig S13B).

Trafficking and secretion of Wnt5a are also dependent on WLS
(Das et al, 2012). Wnt5a is widely involved in activating noncanonical
Wnt/Ca²⁺ or Wnt/planar cell polarity pathways and also in regulating
canonical Wnt signaling pathway (Guo & Xing, 2022). As mentioned
above, the addition of Wnt3a or treatment with CHIR99021 resulted in
a reduced number of actin patches in knockout cells. We therefore
assessed whether Wnt5a could also similarly rescue the actin patch
phenotype. The addition of Wnt5a neither reduced patches in *Clta⁻/⁻*
and *Clta_b⁻/⁻* mESCs nor in *Clta⁻/⁻* and *Clta_b⁻/⁻* L cells (Fig S14A and
D). However, the total amount of Wnt5a was significantly down-
regulated in *Clta⁻/⁻* mESCs (Fig S14B and C).

To further explore the cross-talk between Wnt signaling and
actin organization, we sought to determine whether a reorganized
actin cytoskeleton could also alter Wnt signaling. We treated WT
mESCs with latrunculin A (Lat A) (a pharmacological inhibitor of
actin polymerization) and examined β-catenin transcriptional ac-
tivity using the TOPFlash assay. β-Catenin transcriptional activity
was reduced in WT and *Cltb⁻/⁻* mESCs treated with Lat A com-
pared with their DMSO-treated counterparts. However, *Clta⁻/⁻* and
*Clta_b⁻/⁻* mESCs treated with Lat A showed similarly reduced
levels of β-catenin transcriptional activity compared with their
control (Fig 5F). This indicated an involvement of actin organization
in Wnt signaling activation. In addition to reduced β-catenin
transcriptional activity upon Lat A treatment, Wnt3a secretion in WT
and CLC KO Wnt3a L cells was also found to be reduced under these
conditions (Fig 5G).

As mentioned previously, reduced levels of Hip1R caused actin
patches in *Clta⁻/⁻* and *Clta_b⁻/⁻* mESCs, with CHIR99021 treatment
resulting in a reduction of actin patches. We therefore asked
whether treatment of *Clta⁻/⁻* and *Clta_b⁻/⁻* mESCs with CHIR99021
could rescue Hip1R protein levels. Upon treatment with CHIR99021,
*Clta⁻/⁻* and *Clta_b⁻/⁻* mESCs showed restoration of protein levels of
Hip1R (Fig 5H and I). Together, our results demonstrate the presence
of a pathway wherein CLCa regulates Wnt secretion, and through
activation of the downstream Wnt signaling pathway, Hip1R levels
are maintained, resulting in regulated actin organization in mESCs.
CLC-mediated regulation thus provides an additional layer of
regulation of the Wnt pathway and actin organization, offering
regulatory checkpoints for developmental decisions.

## Discussion

Regulation of Wnt signaling and accurate organization of the actin
cytoskeleton are critical during early embryonic development
(Logan & Nusse, 2004; Clevers, 2006; Lim & Plachta, 2021). Our
study shows a unique involvement of CLCs in regulating Wnt
secretion in a manner involving actin organization. We describe a
role of CLCa in the trafficking of Wnt3a and its secretion mediator,
WLS, from the Golgi to the plasma membrane. Previous reports
studying the secretion of Wnt via an exosome-mediated pathway
have also demonstrated the presence of clathrin and certain
actin-interacting proteins in Wnt-containing exosomes (Gross
et al, 2012).

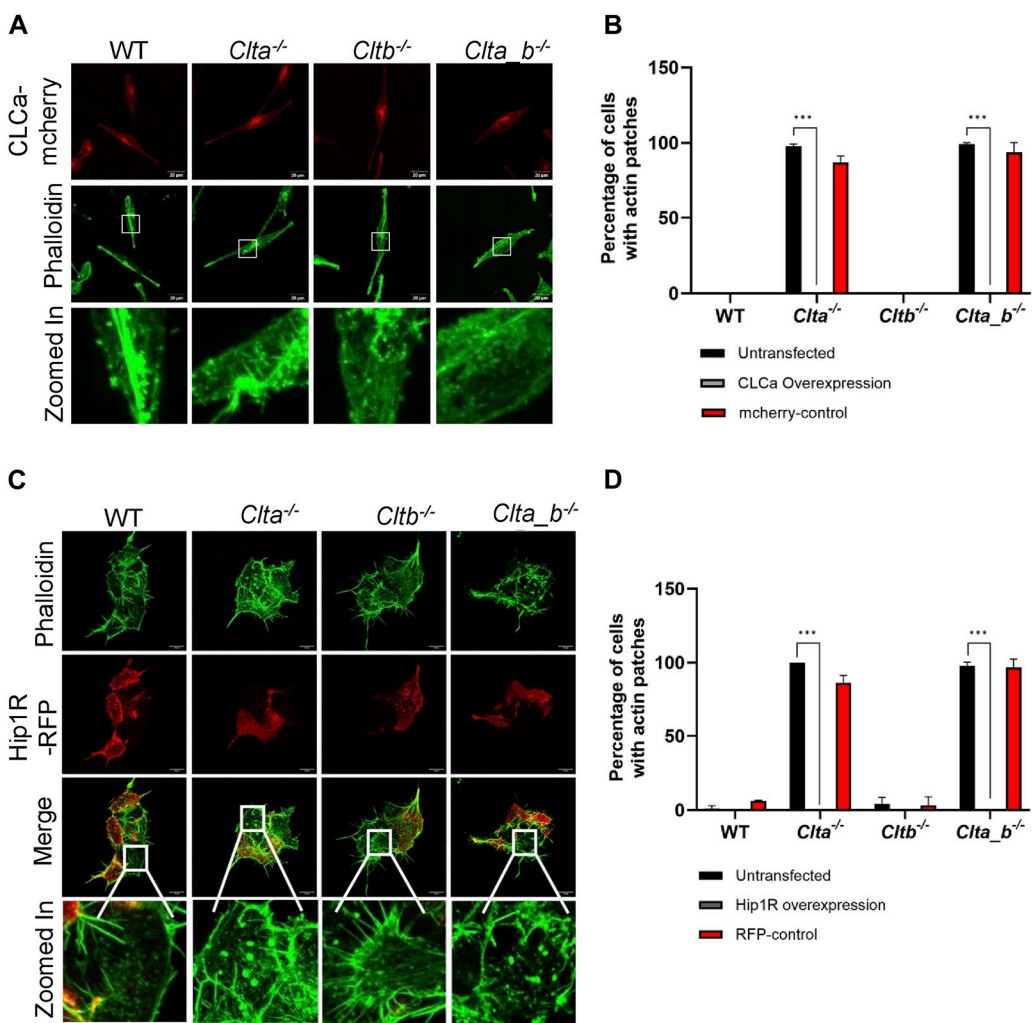

**Figure 4. CLCa regulates actin organization via Hip1R.**
**(A)** Confocal images showing phalloidin staining for F-actin in WT, *Clta⁻/⁻*, *Cltb⁻/⁻*, and *Clta_b⁻/⁻* L cells transfected with CLCa-mCherry (scale bar—20 μm). **(B)** Bar graph showing quantitation of cells having actin patches upon the overexpression of CLCa-mCherry or mCherry empty vector control plasmids, compared with untransfected cells. **(C)** Confocal images showing phalloidin staining for F-actin in WT, *Clta⁻/⁻*, *Cltb⁻/⁻*, and *Clta_b⁻/⁻* mESCs transfected with Hip1R-RFP (scale bar—10 μm). **(D)** Bar graph showing quantitation of cells having actin patches upon the overexpression of Hip1R-RFP or RFP empty vector control plasmids, compared with untransfected cells. Error bars represent the mean ± SEM for experiments in triplicates (N = 3). *P < 0.05; **P < 0.01; ***P < 0.001 by t test. Source data are available for this figure.

We also observe a unique cross-talk between Wnt signaling and actin organization, as treatment of cells with either exogenous Wnt or pharmacological inhibition of GSK3β resulted in the dissolution of actin patches, albeit in a cell type–dependent manner. In mESCs, the addition of CHIR99021 resulted in reduction of actin patches in *Clta⁻/⁻* and *Clta_b⁻/⁻* mESCs. However, long-term treatment with Wnt3a did not support the survival of mESCs. On the other hand, treatment of L cells with either CHIR99021 or recombinant Wnt3a resulted in the reduction of actin patches in *Clta⁻/⁻* L cells but not in *Clta_b⁻/⁻* L cells, suggesting that the requirement of CLC and Wnt signaling in actin organization may be cell type–specific.

GSK3β is known to regulate actin filament organization and focal adhesion by regulating Arp2/3 activity via Rac1 activation in rat fibroblasts (To et al, 2017). GSK3β is also involved in filopodium formation by regulating the activity of Rac1 and Cdc42 (Etienne-Manneville & Hall, 2003; Jacquemet et al, 2015). GSK3β activity is required for the formation of invadopodia by regulating MT1-MMP expression. Inhibition of GSK3β activity with AR-A014418 (a GSK-3 inhibitor) caused reduction in MT1-MMP expression and further reduction in the number of invadopodia (Chikano et al, 2015). Focal adhesion proteins such as FAK and paxillin are direct targets for phosphorylation by GSK3β. Inhibition of GSK3β activity resulted in reduced phosphorylation of FAK and paxillin and cell motility (Xu et al, 2014; Mai et al, 2021). All these studies implicate the involvement of GSK3β in rearrangement of the actin cytoskeleton. In our study, we also observe that the inhibition of GSK3β activity resulted in a reduction in the formation of actin patches in *Clta⁻/⁻* and *Clta_b⁻/⁻* cells. Whether this is through the modulation of proteins such as Rac1, Cdc42, MMPs, and other focal adhesion proteins will form the basis of future studies.

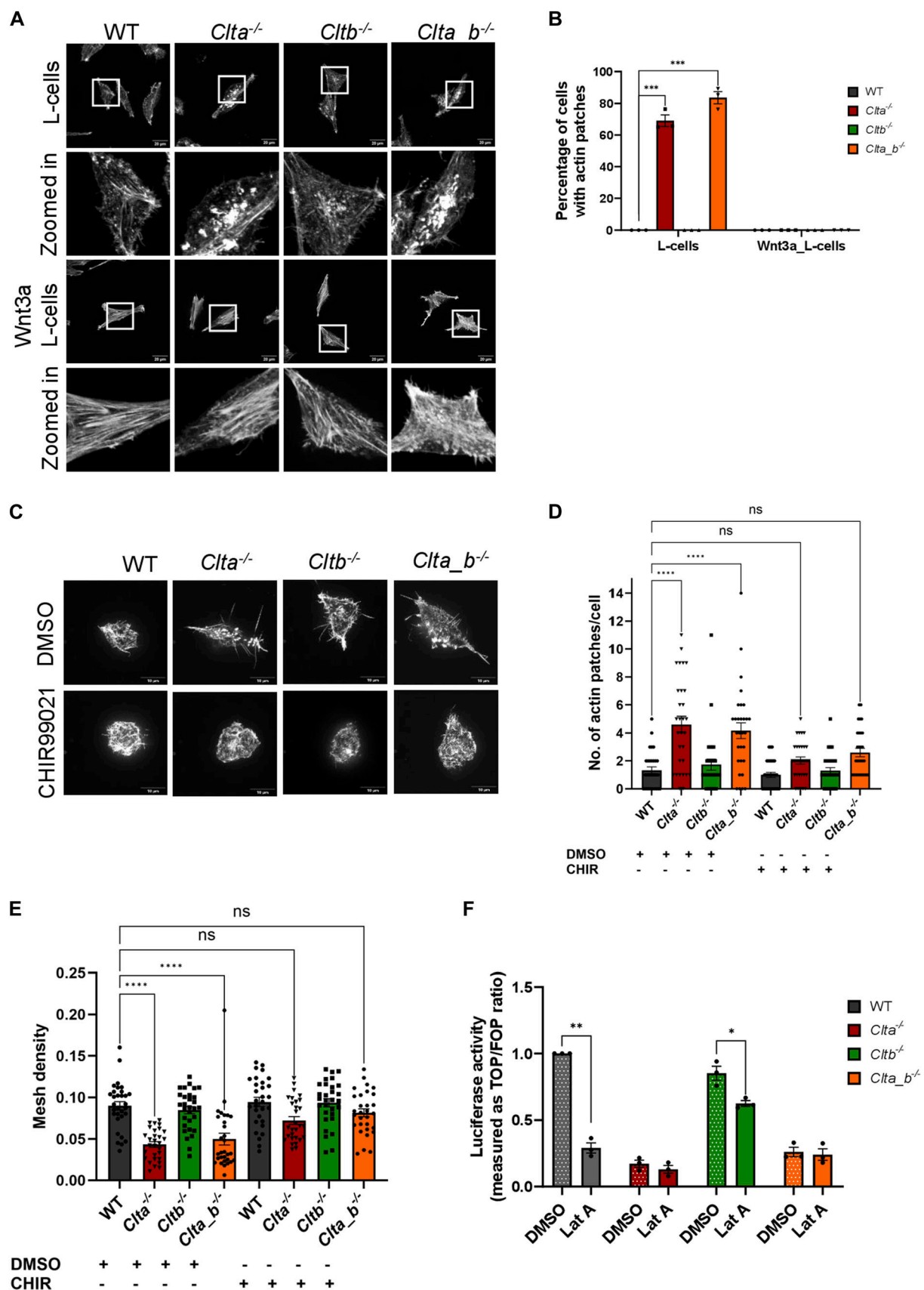

**Figure 5.**
(Continued)

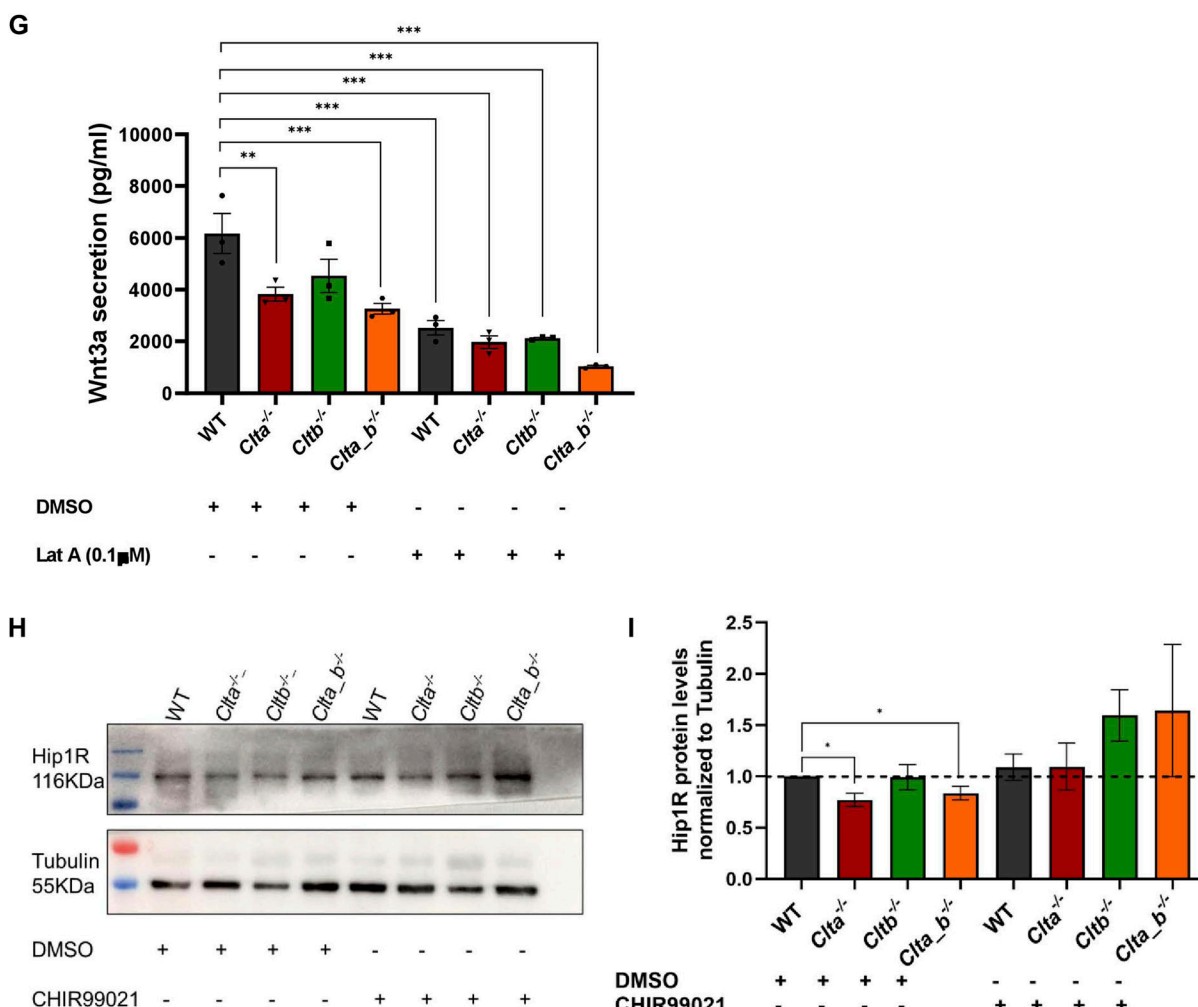

**Figure 5. Secretion of Wnt3a is dependent on CLCa and an organized actin network.**
**(A)** Super-resolution spinning disk confocal microscopic images depicting F-actin stained with phalloidin in WT, *Clta⁻/⁻*, *Cltb⁻/⁻*, and *Clta_b⁻/⁻* L cells and Wnt3a L cells. Scale bar—20 μm. **(B)** Quantitation of cells showing actin patches and WT, *Clta⁻/⁻*, *Cltb⁻/⁻*, and *Clta_b⁻/⁻* L cells and Wnt3a L cells. **(C)** Super-resolution spinning disk confocal microscopic images showing F-actin stained with phalloidin in WT, *Clta⁻/⁻*, *Cltb⁻/⁻*, and *Clta_b⁻/⁻* mESCs treated with DMSO or CHIR99021 (6 μM). Scale bar—10 μm. **(D, E)** Bar graphs showing quantitative analysis of various features of actin structures: (D) actin patches; and (E) mesh density in WT, *Clta⁻/⁻*, *Cltb⁻/⁻*, and *Clta_b⁻/⁻* mESCs treated with DMSO or 6 μM CHIR99021 (CHIR). **(F)** TOPFlash reporter assay showing levels of β-catenin–mediated transcriptional activation in WT, *Clta⁻/⁻*, *Cltb⁻/⁻*, and *Clta_b⁻/⁻* mESCs, treated with DMSO or 0.2 μM latrunculin A (Lat A). **(G)** Quantitation of secreted Wnt3a from WT, *Clta⁻/⁻*, *Cltb⁻/⁻*, and *Clta_b⁻/⁻* Wnt3a L cells upon treatment with latrunculin A for 6 h. Error bars represent the mean ± SEM for experiments in triplicates (N = 3). *P < 0.05; **P < 0.01; ***P < 0.001 by one-way ANOVA test. **(H)** Western blots showing total protein levels of Hip1R and tubulin in WT, *Clta⁻/⁻*, *Cltb⁻/⁻*, and *Clta_b⁻/⁻* mESCs, treated with DMSO or CHIR99021 (6 μM). **(I)** Bar graph showing total protein levels of Hip1R normalized to tubulin for WT, *Clta⁻/⁻*, *Cltb⁻/⁻*, and *Clta_b⁻/⁻* mESCs treated with DMSO or CHIR99021 (6 μM). Error bars represent the mean ± SEM for experiments in triplicates (N = 3). *P < 0.05; **P < 0.01; ***P < 0.001 by *t* test.
Source data are available for this figure.

We also observe that the loss of CLCa, specifically, results in the reorganization of actin toward the basal surface of the cell into patches with enrichment of components of the Arp2/3 complex and cortactin. The Arp2/3 complex is involved in the growth of branched actin filaments, and cortactin plays a role in stabilizing actin filaments by interacting with the Arp2/3 complex and actin filaments through its N terminus. Their enrichment into actin patches suggests a loss or reduction of available functional molecules in the cell. Furthermore, the appearance of patches in cells preferentially lacking CLCa is suggestive of its involvement in the trafficking and/ or recycling of these proteins at the basal surfaces of cells. At the site of endocytosis, CLCs serve as a link between endocytosis and actin via Hip1/Hip1R. Hip1R brings actin to the CCP to induce membrane invagination. Stabilization and branching of actin filaments are achieved by cortactin and Arp2/3 complex, respectively. Loss of CLCa has also been shown to result in the disrupted interaction between actin and actin binding proteins such as Hip1R and cortactin (Le Clainche et al, 2007), causing the over-assembly of actin at sites of endocytosis.

Previous reports show that siRNA-mediated knockdown of both CLCs in HeLa cells resulted in actin patches along with reduced Hip1R levels (Poupon et al, 2008). However, the mechanisms of formation of these patches appear to be different from what we observe in terms of involvement of Hip1R, indicating that this may

differ in a cell type–specific manner. Similarly, astrocytes showed the presence of actin patches upon loss of either CLC (Mukehirn et al, 2021). However, our results point strongly toward the function of only CLCa. Furthermore, the patches observed by Mukenhirn et al morphologically differ from what we observe in mESCs, further raising the possibility that these may differ in their molecular composition. In addition, we demonstrate for the first time that the reduction in Hip1R protein levels observed upon loss of CLCa can be restored through Wnt pathway activation.

The actin cytoskeleton is crucial for proper embryonic development by regulating cell shape and dynamics (Lim & Plachta, 2021). Alteration in the availability and localization of actin binding proteins such as Arp2/3, Hip1R, and cortactin can result in change of cell shape, mechanics, and gene expression. This could further lead to changes in cell fate specification resulting in developmental defects and disorders.

Various studies have also shown that actin assembly, along with Arp2/3 and cortactin, is also involved in Golgi and endocytic trafficking (Cao et al, 2005; Steffen et al, 2008; Salvarezza et al, 2009; Sung et al, 2011). Perturbation of actin assembly by actin targeting drugs or defects in recruitment of actin to the Golgi results in compaction and fragmentation of Golgi. Proper actin assembly and organization at the Golgi are required for maintaining cisternal ultrastructure, intraluminal pH, and vesicular trafficking (Valderrama et al, 1998, 2001; Egea et al, 2006; Lázaro-Diéguez et al, 2006; Campellone et al, 2008). Actin is also involved in maintaining the pH gradient across the Golgi by regulating the activity of vacuolar H+-ATPase (V-ATPase) from the cis-Golgi to the TGN. Depolymerization of actin results in loss of activity of V-ATPase, causing further imbalance in the pH gradient across the Golgi (Casey et al, 2010; Serra-Peinado et al, 2016).

Knockdown of cortactin in cells also resulted in a compact Golgi morphology along with defective cholesterol trafficking at the late endosomal/lysosomal (LE/Lys) compartments. However, these cells did not show any defect in the transport of VSV-G from the Golgi to the PM (Kirkbride et al, 2012). Interestingly, a dominant negative mutant of cortactin showed reduction in cell-surface VSV-G protein (Cao et al, 2005). Cortactin knockdown HT1080 cells showed defective fibronectin (FN) secretion resulting in altered cell motility and lamellipodium movement (Sung et al, 2011). Budding of clathrin-coated vesicles from the TGN also requires actin polymerization. Clathrin molecules recruit WAVE complex and components of the Arp2/3 complex to provide the force for budding from the TGN (Anitei et al, 2010). Perturbation of these actin binding factors causes reduced CCV assemblies at the TGN and defective transport to lysosomes (Anitei et al, 2010; Chakrabarti et al, 2021). Previous studies also demonstrated that CLC depletion or Hip1R depletion altered actin assembly near the TGN, affecting trafficking of CI-MPR and cathepsin D maturation (Engqvist-Goldstein et al, 2004; Poupon et al, 2008).

Our study also reveals a role of specific CLCs during early development. In vitro differentiation of *Cltb*$^{-/-}$ KO mESCs showed a preference toward the endodermal lineage and also toward cardiac differentiation. Previous observations by Chen et al (2017) also show that the individual expression of only CLCb results in altered activation of components of the Wnt signaling pathway. Furthermore,

activation of canonical Wnt signaling is involved in the induction of primitive endoderm in mESCs (Price et al, 2013), further indicating a potential role of specific CLCs in regulating developmental decisions.

Previous reports have also demonstrated a cell type–specific expression of CLCs, with B cells exclusively expressing CLCa, whereas levels of CLCb were significantly higher in migratory trophoblast cells (Majeed et al, 2014). The essential functions of both the Wnt pathway and an organized cytoskeleton during early developmental decisions are already well elucidated. Regulating the trafficking of individual components of these two processes through the involvement of specific CLCs provides an additional layer of control to ensure accurate spatial and temporal decisions during development. Understanding whether individual CLCs function at distinct stages of development to facilitate transport of specific regulatory proteins is therefore of tremendous importance and will provide further detail to the complex picture of early development.

# Materials and Methods

### Cell culture

V6.5 mESCs were plated on 0.2% gelatin-coated tissue culture plates (Corning) and cultured in knockout DMEM (Gibco) supplemented with 15% FBS (Gibco), 0.1 mM beta-mercaptoethanol, 2 mM L-glutamine, 0.1 mM nonessential amino acids, 5,000 U/ml penicillin/streptomycin, and leukemia inhibitory factor (LIF) (ES+LIF medium). mESCs were incubated at 37°C and 5% (vol/vol) $CO_2$ incubator. mESCs were treated with CHIR99021, a GSK-3 inhibitor at a final concentration of 6 $\mu$M in ES+LIF medium.

Mouse L cells and Wnt3a L cells were grown on cell culture plates (Corning) in DMEM (Gibco) containing 10% FBS, 1X penicillin/ streptomycin. Cells were grown in 37°C and 5% (vol/vol) $CO_2$ incubator.

### Generation of knockout lines using CRISPR-Cas9

Single-guide RNAs (sgRNAs) were designed to target exons 2 and 3 of the *Clta* gene, and exon 1 of the *Cltb* gene and cloned into the pX459 vector (Addgene). (Table S1). For single CLC KO, 2 $\mu$g of the sgRNA-containing vector was transfected into 50 × 10$^3$ mESCs plated in a six-well plate using Lipofectamine 2000 as per the manufacturer's instructions (Invitrogen). For CLC double KO, 1.5 $\mu$g of each sgRNA-containing vector was transfected. 24 h post-transfection, puromycin was added to the media for selection at a concentration of 1 $\mu$g/ul for 48 h. For L cells, puromycin selection was carried out at a concentration of 7.5 $\mu$g/$\mu$l for 72 h. Post-antibiotic selection, cells were plated in 96-well tissue culture plates (Corning) at single-cell density per well. After 5 d, screening of KO clones was done using a dot blot, by placing 5 $\mu$l of protein lysate from a single cell–derived colony onto methanol-treated wet polyvinylidene fluoride (PVDF) membrane. After blocking, the membrane was incubated with

primary antibody, followed by incubation with the appropriate secondary antibody and development.

## Transfection

20,000 cells were seeded on 12-mm coverslips in a 24-well plate overnight. The next day before transfection, medium was replaced with fresh medium. 1 µg of the desired plasmid was transfected using Lipofectamine 2000 reagent as per the manufacturer's instructions. After incubation, transfection mixture was added dropwise to the cells. After 6 h, the media were replaced with fresh medium and cells were kept in the incubator at 37°C with 5% CO2. Plasmids used were Hip1R-tdimer-RFP-N1 (plasmid # 27700; Addgene), CLCa-mCherry, mannosidase II-GFP, and WLS-mCherry.

## Embryoid body formation

Embryoid bodies (EBs) were formed using the hanging drop method (500 cells/ 20 ml drop) in ES media without LIF. After 4 d, EBs were transferred onto 0.2% gelatin-coated dishes and cultured for 7 d in DMEM containing 10% FBS, 0.1 mM beta-mercaptoethanol, 2 mM L-glutamine, 0.1 mM nonessential amino acids, 5,000 U/ml penicillin/streptomycin (10% FBS medium). For recombinant Wnt3a treatment, EBs plated on gelatin-coated dishes were treated with 100 ng Wnt3a every alternate day in 10% FBS media for 7 d. Post-7 d, cells were collected for RNA isolation. For immunocytochemistry, EBs were plated on 0.2% gelatin-coated 12-mm coverslips and cultured in 10% FBS medium for 7 d and fixed with 4% PFA for immunocytochemistry.

## Directed neuronal differentiation of mESCs

To differentiate mESCs into neurons, embryoid bodies were formed using the hanging drop method. 500 cells were plated in 20 µl drops as hanging drops for four days in ES media without LIF. These were transferred onto plates coated with polyornithine (Sigma-Aldrich)- and laminin (Roche)-containing DMEM (Invitrogen) supplemented with 10% FBS (Invitrogen), 5 µm retinoic acid, 2 mM L-glutamine (Invitrogen), 1X nonessential amino acids (Invitrogen), 500 µl beta-mercaptoethanol (Invitrogen), and 1% vol/vol penicillin/streptomycin (Invitrogen) per 100 ml media for 4 d with medium change after 2 d. The media were then replaced with DMEM containing 1X B27 supplement (Invitrogen), 2 mM L-glutamine, 1X nonessential amino acids, 500 µl beta-mercaptoethanol, and 1% vol/vol penicillin/streptomycin per 100 ml media for 5 d with medium change every 2 d. The differentiated cells were then harvested in TRIzol for RNA isolation.

## Western blotting

Protein lysates were prepared using RIPA buffer containing protease inhibitor cocktail and PMSF. Lysates were kept for 20 min on ice for lysis followed by centrifugation at 15,000$g$ for 20 min at 4°C. The protein concentration of the supernatant was determined using Bradford's reagent. An equal concentration of total protein lysates was loaded onto SDS–PAGE gel and separated by electrophoresis,

followed by transfer onto a PVDF membrane. After transfer, the membrane was blocked in 5% BSA at RT for 1 h. Post-blocking, the membrane was incubated with the appropriate primary antibody overnight at 4°C with gentle rocking. The next day, the blot was washed twice for 10 min using Tris-buffered saline (1X TBS) containing 0.1% Tween-20 (TBS-T). Membranes were incubated with an HRP-conjugated secondary antibody (1:1,000) for 1 h at room temperature. Thermo Fisher Scientific SuperSignal West Femto Maximum Sensitivity Substrate reagent was added to the membranes, and images were captured post-exposure using a ChemiDoc system (AI600; GE Healthcare). Western blot images were quantified using ImageJ software.

## qRT–PCR

Total RNA was isolated using TRIzol (Invitrogen) and quantified using NanoDrop Spectrophotometer (Thermo Fisher Scientific). For mRNA amplification, 1 µg RNA was treated with DNase I (Invitrogen) and reverse-transcribed using a Verso cDNA synthesis kit (Invitrogen) using random hexamers. The total cDNA obtained was diluted 1:10 and used as a template with appropriate primers (Table S2). Applied Biosystems SYBR Green Master Mix (Cat. No. A25742) was used to set up qRT–PCR. mRNA expression was normalized to RPL7 or GAPDH and represented relative to the control sample.

## Transferrin uptake assay

WT mESCs and CLC KO mESCs were plated on 0.2% gelatin-coated 12-mm glass coverslips. After 24 h of plating, cells were serum-starved in knockout DMEM at 37°C. After 1 h of serum starvation, cells were incubated with Alexa Fluor 488–labeled transferrin (10 µg/ml) for 15 min at 37°C, followed by washing with acid wash buffer for 30 s to remove nonspecifically bound transferrin. Cells were then washed three times in PBS, fixed in 4% PFA for 15 min, and imaged using the confocal microscope. All images were analyzed using ImageJ software, and corrected total cell fluorescence (CTCF) was calculated for each cell type from three independent experiments.

## Immunocytochemistry

mESCs were plated on 0.2% gelatin-coated 12-mm glass coverslips and fixed after 6 h of plating with 4% PFA for 15 min at RT, followed by washing with 1XPBS. For L cells and Wnt3a L cells, cells were plated on 12-mm glass coverslips and fixed after 24 of plating. Fixed cells were permeabilized with 0.2% Triton X-100 for 10 min at RT. Blocking was done for 1 h with 5% BSA, followed by primary antibody incubation overnight at 4°C. After primary antibody incubation, cells were washed with 1xPBST three times for 5 min each. Cells were then incubated with the appropriate secondary antibody for 1 h at RT, followed by washing with 1xPBST three times. Coverslips were mounted using VECTASHIELD, and imaging was done on an Olympus FV3000 confocal microscope. A list of antibodies used is provided in Table S3.

**ELISA for secretion of Wnt3a from Wnt3a L cells**

50,000 Wnt3a L cells were plated in a 12-well plate. After 12 h of plating, cells were serum-starved by maintaining them in non-supplemented DMEM for 12 h. Fresh medium was added and collected at different time points such as 6, 12, 24, and 48 h. ELISA (Wnt3a DuoSet ELISA R&D Systems) was performed using the kit instructions. Absorbance was determined at 450 and 540 nm using the plate reader from Thermo Fisher Scientific.

**TIRF microscopy**

10,000 WT and CLC KO Wnt3a L cells were plated in 15-mm glass-bottom confocal dishes. Cells were starved in nonsupplemented DMEM for 12 h after 12 h of seeding. After starvation, fresh medium was added for 6 h, followed by fixation with 4% PFA and immuno-staining for Wnt3a. Cells were imaged using the Nikon Total Internal Reflection Fluorescence (TIRF) Inverted Microscope (Eclipse Ti) with 100x objective.

**RNA sequencing**

*RNA extraction and library preparation*
For library construction, total RNA was extracted, and mRNA was purified using oligo-dT beads (TruSeq RNA Sample Preparation Kit; Illumina) from 1 µg of intact total RNA. RNA purity was checked using the Bioanalyzer 2100 system (Agilent Technologies), and RNA with an RIN value of 7.0 and above was used for library preparation. The purified mRNA was fragmented at 90°C in the presence of divalent cations. The fragments were reverse-transcribed using random hexamers and Superscript II Reverse Transcriptase (Life Technologies). Second strand cDNA was synthesized on the first strand template using RNase and DNA polymerase I. The cDNAs so obtained were cleaned using Beckman Coulter SPRI beads.

Sequencing libraries were generated using TruSeq RNA Library Prep Kits for Illumina (NEB) according to the manufacturer's instructions, and index codes were added to attribute sequences to each sample. Clustering of the index-coded samples was performed on cBot Cluster Generation System using the TruSeq PE Cluster Kit v3-cBot-HS (Illumina) according to the manufacturer's recommendations. Finally, paired-end reads of 151 bp were generated via an Illumina MiSeq platform.

*Sequencing and quality filtering*
Raw reads were checked for different quality parameters including phred score, length, contamination, GC%. Depending on raw data QC, we used the Fastp filtration tool to filter out high-quality reads from raw data. Fastp is an ultrafast FASTQ processor with functions for quality control and data filtering. It performs quality control, adapter trimming, quality filtering, per-read quality pruning, and many other operations with a single scan of the FASTQ data. A cutoff of 30 was set for the quality phred score, and only high-quality reads were retained and a minimum read length cutoff was kept at 70. High-quality filtered reads from all the samples were used to align reads to the reference genome by HISAT2 with default parameters.

*Differential gene expression analysis*
DESeq2 was used to perform differential analysis between read counts at a gene level for the alternative conditions. The total sum of read counts for each gene is taken as at least 10 among the libraries under comparison. The selected genes were taken further for differential expression analysis. A *P*-value cutoff of 0.05 and less was used to identify the significantly expressed genes, and a $\log_2$ fold change cutoff of (+2) and higher for "up-regulated" and (−2) and less for "down-regulated" genes was used. Genes that are significant but below the set $\log_2$ fold change cutoff are labeled as "baseline." Genes with a higher *P*-value than set cutoff or NA are labeled as "Not significant."

**Image analysis**

- For the analysis on actin patches, the first five bottom stacks were Z-projected with average intensity in ImageJ. Further analysis was done using custom-written scripts in MATLAB, where an adaptive thresholding method was used with a sensitivity of 0.2. For distinguishing actin patches/puncta from long actin protrusions, an eccentricity cutoff of 0.88 was used. For actin meshwork analysis, ridge detection was performed on Fiji with a minimum length of four pixels and a maximum length of 30. The ridge detection was then converted into a distance map. The ridge detection was also inverted and subjected to ultimate erosion giving the center of each pore. The data from the distance map image and ultimate erosion image obtained from the original ridge detection image were multiplied to give pore radii as pixel values. Mesh density was calculated by measuring a total length of the actin mesh per unit area.
- WLS intensity calculations within the Golgi

  All images were average intensity–projected for all the Z-stacks in ImageJ. The region of interest was marked using GM130 to stain the Golgi, after which WLS intensity was measured within the ROI. CTCF was calculated after background subtraction.

**Statistical analysis**

We performed *t* test for all experiments involving Western blotting, qRT-PCR, luciferase assay, and CTCF measurements. Error bars represent the mean ± SEM for experiments in triplicates (N = 3).

For ELISA and CTCF measurements, a one-way ANOVA test was performed. Error bars represent the mean ± SEM for experiments in triplicates (N = 3).

# Supplementary Information

# Acknowledgements

This work was supported by funds to D Subramanyam from the Department of Biotechnology (DBT) BT/PR30450/Med/31/399/2018. M

Tiwari and J Das are supported by Senior Research Fellowships from UGC, India. We thank Prof. Pete Cullen for the WLS-mCherry construct obtained through Dr. Maddika Subba Reddy. We also thank Prof. Christien Merrifield and Dr. Thomas Pucadyil for Hip1R-tdimer-RFP-N1 and CLCa-mCherry plasmids, respectively. We thank members of the Subramanyam laboratory for constructive discussion. D Subramanyam thanks AVR for support.

## Author Contributions

M Tiwari: conceptualization, data curation, formal analysis, validation, investigation, visualization, methodology, and writing—original draft, review, and editing.
M Dingankar: data curation, formal analysis, investigation, methodology, and writing—review and editing.
J Das: data curation, formal analysis, investigation, visualization, methodology, and writing—review and editing.
SS R: data curation, formal analysis, investigation, and writing—review and editing.
A Solanki: investigation, methodology, and writing—review and editing.
D Subramanyam: conceptualization, resources, data curation, software, supervision, funding acquisition, validation, investigation, visualization, project administration, and writing—original draft, review, and editing.

## Conflict of Interest Statement

The authors declare that they have no conflict of interest.

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
