## [Reviewer comments · Life Science Alliance]

Life Science Alliance

CLCa mediates a novel cross-talk between Wnt secretion and actin organization

Mahak Tiwari, Mihir Dingankar, Jyoti Das, Sreelekshmi SR, Apurv Solanki, and Deepa Subramanyam

DOI: <https://doi.org/10.26508/lsa.202402962>

Corresponding author(s): Deepa Subramanyam, National Centre for Cell Science

Review Timeline:

Submission Date:	2024-07-27
Editorial Decision:	2024-09-09
Revision Received:	2025-03-02
Editorial Decision:	2025-03-20
Revision Received:	2025-04-17
Accepted:	2025-04-17

Scientific Editor: Tim Fessenden

Transaction Report:

September 9, 2024

Re: Life Science Alliance manuscript #LSA-2024-02962-T

Dr. Deepa Subramanyam
National Centre for Cell Science
SP Pune University
Ganeshkhind Road
Pune, Maharashtra 411007
India

Dear Dr. Subramanyam,

Thank you for submitting your manuscript entitled "CLCa mediates a novel cross-talk between Wnt secretion and actin organization" to Life Science Alliance. The manuscript was assessed by expert reviewers, whose comments are appended to this letter. We invite you to submit a revised manuscript addressing the Reviewer comments.

Thank you for this interesting contribution to Life Science Alliance. We are looking forward to receiving your revised manuscript.

Sincerely,

B. MANUSCRIPT ORGANIZATION AND FORMATTING:

Reviewer #1 (Comments to the Authors (Required)):

Despite extensive work, there are still many open questions on signaling pathways in early development of murine embryonic stem cells (mESCs). In this context, the authors found a novel mechanism based on mammalian clathrin light chains (CLSa, CLCb), which are thought to be critical players in early signaling involving the Wnt pathway and the actin cytoskeleton organization. In particular, during CLC knockout in mESCs, they found that Wnt signaling is significantly altered due to impaired trafficking of its coreceptor, WLS. The reduced Wnt signaling leads to lower Hip1R and consequently to disorganization of actin in the cytoskeleton. This mechanism is certainly interesting, but it stands more or less alone and the reader would like to have a better embedding of this result in other signaling pathways, probably also regulating cytoskeletal organization. Several points that need clarification:

1. Fig. 1 shows a transcriptome analysis comparing wild-type and CLC knockout mESCs. This demonstrates that many more genes are altered in their transcription level. Therefore, more evidence should be given why the authors only focused on the mechanism shown here. It is likely that other mechanism due to CLCa knockdown are also relevant for cell development and in particular for actin organization.
2. Unfortunately, the paper is rather descriptive and contains only some immunohistochemical pictures for the altered actin organization. Additional functional data would be useful.
3. It is well known that cardiac development is critically dependent on actin organization. Therefore, it would be interesting to see whether beating cardiomyocytes are still developed from CLCa knockout mESCs.
4. If beating cardiomyocytes are developed it would be interesting to see whether there is signaling, e.g. modulation of ionic channels.
5. Similar questions are also relevant for neuronal or skeleton muscle differentiation where actin organization is critical.
6. Further rescue experiments should be performed. It would be interesting to study whether Wnt application can rescue the defect or whether other factors are still important to rescue functional development of cells from knockout mESCs.
7. For a better understanding of the different pathways involved in early mESCs development, a schematic demonstration would be helpful for the reader who wants to see how the found effects are embedded in other signaling cascades.

Reviewer #2 (Comments to the Authors (Required)):

In this manuscript, the authors describe experiments that link the clathrin light chain (CLC) proteins to Wnt trafficking and actin organization. By generating knockouts (KO) in mouse ESCs and in L-cells, they demonstrate that CLC proteins influence Wnt signaling. Expression of multiple Wnt target genes is downregulated in Clt KO lines, suggesting that impaired trafficking of Wnt3a leads to less downstream signaling, an effect that can be rescued by exogenous Wnt pathway activation. Using cells that overexpress and secrete Wnt3a (L-Wnt3a cells), they demonstrate that Wnt secretion is diminished in Clt KO lines. Since Wnt signaling has been linked to actin organization, the authors extend their studies to examine the effect of Clt KO on actin fibers. While there may be a direct link between Clts, Wnt and actin, this part of the paper is less convincing.

An overarching critique of this work is that it relies heavily on the use of clonal cell lines, both for the mESC and the L-cell studies. Clonal variations unrelated to the specific alteration/mutation (Clt knockout or Wnt3a overexpression) may account for some, many or even all observed phenotypes. This is a common caveat in such studies and it is difficult to circumnavigate without analyzing multiple clones per genetic alteration. One possible way to address this critique is to use inducible systems. In the case of the Clta/b genes, an inducible shRNA knockdown system could be employed. For Wnt3a, an inducible overexpression system could be used. Such systems exist and should be incorporated into these studies.

Several experiments that examine Wnt3a trafficking would benefit from the use of a Porcn inhibitor as a control for Wnt secretion. Porcn inhibition (nearly) completely blocks Wnt secretion and observed effects may be more pronounced than what is observed in Clt Kos. This would be useful in assessing the extent of Wnt secretion blockage in the Clt KO lines. For example, in Figure 5a, it would be interesting to observe whether Porcn inhibition can reverse the actin organization of L-Wnt3a cells to that of L-cells. Wash-out of Porcn inhibitor could be employed as a means to do "pulse-chase" experiments.

The loss of CLCs may also impact secretion and recycling of Wnt receptors, such as the Frizzleds and Lrp5/6. The fact that Clt knockout lines can still respond to exogenous Wnt3a treatment (Figure 1f) indicates that receptors are present and capable of transducing Wnt signals. Nonetheless, this should be examined.

Multiple experiments indicate that Clta plays a bigger role in Wnt trafficking than Cltb (e.g. Figures 1d, 2e,f, 3a-d), an observation that the authors point out but fail to pursue further. It would be interesting to test whether overexpression of Clta only can rescue the phenotypes observed in Clta_b double KO lines. The Discussion provides very little commentary on the distinct roles of these 2 genes in Wnt signaling and development.

In EBs, expression of pluripotency markers is elevated in Clta knockout cells, but not in Cltb or Clta_b. How do the authors explain that the increased levels observed in the single Clta KO line is not observed in the Clta_b double KO line? The same question applies to the elevated levels of differentiation markers (T, GATA4, AFP) in Cltb KO EBs: how is that this effect is not observed in the double KOs?

Supp Figure 1: Authors claim that there are no significant changes in expression of pluripotency markers between WT and Clt knockout lines, however, Oct4 levels appear to 2-3-fold higher, and in one setting Klf4 expression is elevated 8-fold. Subtle changes such as these may in fact be meaningful. As shown it is difficult to appreciate the statistical significance (p-values?).

Minor:

Figure 1C is confusing. The color-coding of the "Group" on the top of the heatmap does not align with the indicated genotypes at the bottom of the heatmap.

Line 133 and Figure 1D: Dkk1 expression is not reduced in Clta

Line 181-182: what are the time points (6h, 12h, 24h, 48h) relative to? Was media replaced on cells at timepoint 0 and then harvested and quantified for Wnt3a levels at these timepoints? This is not explained well enough in the text or the figure legend.

Although Wnt is in a complex with WLS, WLS is not a Wnt receptor. WLS is a Wnt binding factor required for Wnt secretion. It could also be referred to as a chaperone.

Supp Figure 2: The GO terms are difficult to read. Please increase font sizes.

Line 229: this reference needs to be fixed: (Xia and Kanchanawong, Cell Reports, 2019).

Reviewer #3 (Comments to the Authors (Required)):

This manuscript (LSA-2024-02962-T) investigates the role of Clathrin Light Chains (CLCs), the components of the clathrin triskelion, in the pluripotency and differentiation of embryonic stem cells (ESCs) using mouse ESCs (mESCs). To explore this, the authors generated mESC cell lines with CRISPR-Cas9 genome editing to knock out Clta (encoding CLCa), Cltb (encoding CLCb), and both (Clca/b KO). They characterized these cell lines for the expression of pluripotency and differentiation markers and assessed their endodermal lineage potential through embryoid body formation assays.

Notably, Cltb^{-/-} mESCs displayed a significant upregulation of endoderm markers Gata4 and Gata6 compared to Clta^{-/-} cells, suggesting that CLCs differentially influence developmental cell fate decisions. Further insights into the role of CLCs in mESC pluripotency were obtained through RNA-seq analysis, which revealed dysregulated Wnt/ β -catenin signaling in the absence of CLCs. Specifically, the authors identified decreased expression of Wls, a key regulator of Wnt trafficking from the Golgi to the plasma membrane and secretion.

To explore this further, the authors performed studies using a GSK3 β kinase inhibitor and Wnt3a overexpressing cell lines. These experiments demonstrated that CLCa is crucial for the activation of Wnt3a canonical signaling and Wnt3a secretion. Given previous reports linking CLCs to actin cytoskeletal organization in non-pluripotent cells, the authors investigated whether impaired Wnt3a canonical signaling might contribute to altered actin cytoskeleton organization in the absence of CLCa. Additionally, since GSK3 β is known to regulate actin cytoskeletal organization through Arp2/3, they examined whether Wnt3a canonical signaling regulates actin cytoskeletal organization. Their results showed that Wnt3a and GSK3 β influence the levels of HIP1R, a known regulator of actin cytoskeleton and a ligand of CLCs. Moreover, the use of the actin depolymerizing agent Latrunculin-A revealed that it suppresses Wnt3a signaling in mESCs.

Based on these findings, the authors propose a novel pathway where CLCa regulates Wnt3a secretion, activates downstream Wnt signaling, and influences actin cytoskeletal organization via HIP1R. This study highlights a potential crosstalk between CLCs, Wnt signaling, and actin cytoskeletal organization, which may play a role in developmental decisions.

Overall, the novelty and impact of this study lie in demonstrating the regulation of Wnt3a canonical signaling by CLCs and establishing a crosstalk between CLCs, Wnt3a/ β -catenin signaling, and actin cytoskeletal organization. This is significant for understanding the role of clathrin in actin cytoskeletal biology and pluripotent cell differentiation. While the authors' conclusions are supported by sufficient experimental data, the study could be improved by addressing the following points:

Significant Concerns:

1. The involvement of clathrin in regulating Wnt/ β -catenin signaling is not novel, as previously reported in the following

reference:

o Clathrin regulates Wnt/beta-catenin signaling by affecting Golgi to plasma membrane transport of transmembrane proteins. Munthe E, Raiborg C, Stenmark H, Wenzel EM. *J Cell Sci.* 2020 Jul 9;133(13). doi: 10.1242/jcs.244467.

2. The manuscript does not discuss the status of Wnt5a/b in CLC KO mESCs, which are also regulated by Wls and involved in actin cytoskeletal organization.
3. The importance of CLCa-regulated Wnt3a signaling and actin cytoskeletal organization in pluripotency, differentiation, and fate determination of mESCs is not well elucidated. Additionally, while CLCb appears to regulate endoderm lineage, its role in actin cytoskeletal organization is unclear.
4. The conclusion that Latrunculin-A suppresses Wnt signaling requires additional supporting data. Specifically, does Latrunculin affect Wls or Wnt3a expression in mESCs? Is it acting downstream to GSK3 β ?
5. The preference of Cltb^{-/-} KO mESCs for endodermal lineage is noted, but it is unclear whether this is related to Wnt signaling or actin cytoskeletal organization or both.
6. It would be helpful to clarify whether Wnt5a suppresses the actin patches found in CLCa KO mESCs.
7. The role of HIP1R in actin cytoskeletal organization-whether influenced by its levels alone or also by its activity through phosphorylation or other modifications-requires further clarification.
8. The effect of CLCa on LRP6 expression in mESCs should be explored.
9. Does CLCb bind to HIP1R in mESCs?

Minor Comments:

1. Several references cited in the text are missing from the reference list (e.g., Ferreira et al., 2012; Price et al., 2013; Tsygankova et al., 2019; Xia and Kanchanawong, *Cell Reports*, 2019, among others).
2. In Figure 6, Porcupine is mentioned, but the illustration does not depict or identify PORCN.

We thank the reviewers for their comments and acknowledge and appreciate their suggestions. We have modified our manuscript to address the points that were raised by the reviewers. A point-by-point response to the reviewer's comments is provided below.

Reviewer #1 (Comments to the Authors (Required)):

1. Fig. 1 shows a transcriptome analysis comparing wild-type and CLC knockout mESCs. This demonstrates that many more genes are altered in their transcription level. Therefore, more evidence should be given why the authors only focused on the mechanism shown here. It is likely that other mechanism due to CLCa knockdown are also relevant for cell development and in particular for actin organization.

Response: In the transcriptomic data, KEGG pathway enrichment analysis showed a significant enrichment for Wnt signalling pathway genes as one of the TOP pathways, followed by cell and focal adhesion (Supplementary Fig 3a, 3b and 3c). This is why we focused on the Wnt signalling pathway.

2. Unfortunately, the paper is rather descriptive and contains only some immunohistochemical pictures for the altered actin organization. Additional functional data would be useful.

Response: We thank the reviewer for the comment. We would like to clarify that we have not only provided descriptive measures, but have also provided quantitative and functional data. In the manuscript, we have shown the involvement of CLCa in the organization of actin in a Hip1R-dependent manner and also the requirement of active Wnt signalling pathway in actin cytoskeletal organization by maintaining levels of Hip1R. We also provided quantitative measures for the actin structures such as increased actin patch number, reduced mesh density and increased total area upon the loss of CLCa (Fig. 3b, c, d). We have also provided quantitative measures showing that overexpression of Hip1R results in a rescue of the actin organization (Fig.4); rescue of Wnt secretion upon overexpression of CLCa (Supp Fig 8d,e) and that Hip1R levels are restored upon treatment with CHIR99021 (Fig. 5h,i). In

summary, CLCa appears to be required for the trafficking of Wls, which permits secretion of Wnt3a and activation of a downstream signalling cascade. This then further drives expression of Hip1R which reorganizes actin in a manner that does not permit patch formation.

3. It is well known that cardiac development is critically dependent on actin organization. Therefore, it would be interesting to see whether beating cardiomyocytes are still developed from CLCa knockout mESCs.

Response: The role of canonical Wnt/ β -catenin signaling in cardiac development is well known (Mensa et al., 2024). The timing for activation and inhibition of Wnt signaling during development is crucial for cardiomyocyte differentiation. This includes the early activation of Wnt signaling, promoting the differentiation of ESCs into the mesoderm. The subsequent inhibition of Wnt signaling following mesoderm formation commits mesodermal cells into cardiomyocytes. We did in fact observe that *Clta*^{-/-} mESCs with reduced Wnt signaling activity did not give rise to beating EBs and expression of cardiac markers compared to *Cltb*^{-/-} mESCs. This suggests a differential requirement of CLCa in regulating cardiac differentiation, presumably through the Wnt signalling pathway. These experiments were done at single time point, which is 7 days post plating of EBs on coated dishes. To comment on the involvement of specific CLCs in lineage differentiation requires directed differentiation studies. This is currently beyond the scope of our study. (Supplementary Fig 1e, f)

[Figure removed by editorial staff per authors' request]

4. If beating cardiomyocytes are developed it would be interesting to see whether there is signaling, e.g. modulation of ionic channels- not possible.

Response: *Cltb*^{-/-} mESCs derived EBs showed increased number of beating EBs compared to WT and *Clta*^{-/-} mESCs derived EBs. They also showed increased expression of cardiac genes such *Mlc-2v*, *α-MHC* and *cTnl* compared to WT and *Clta*^{-/-} mESCs derived EBs (Supplementary Fig 1e, f). However, to study the modulation of ion channels is beyond the scope of our study.

5. Similar questions are also relevant for neuronal or skeleton muscle differentiation where actin organization is critical.

Response: We performed directed neuronal differentiation studies with CLC KO mESCs. CLC KO EBs showed decreased expression of neuronal markers such *Map2* and *β-III-Tubulin* compared to WT differentiated EBs, indicating their role in regulating differentiation. (Supplementary Fig 1i).

[Figure removed by editorial staff per authors' request]

6. Further rescue experiments should be performed. It would be interesting to study whether Wnt application can rescue the defect or whether other factors are still important to rescue functional development of cells from knockout mESCs.

Response: We thank the reviewer for this comment. *Clta*^{-/-} mESCs showed downregulated expression of Wnt target genes (Fig1c, 1d) and also showed altered differentiation potential compared to WT mESCs (Supplementary Fig 1g,1h). We

added recombinant Wnt3a to differentiating EBs and performed qPCR to check for the expression of pluripotency and differentiation markers. As mentioned previously, *Clta*^{-/-} mESCs-derived EBs retained the expression of pluripotency markers such as Oct4, Sox2 and Nanog compared to WTmESCs-derived EBs (Fig 1g). However, upon addition of Wnt3a, expression of Oct4 and Sox2 was found to be reduced in differentiated *Clta*^{-/-} mESCs-derived EBs compared to the control *Clta*^{-/-} mESCs-derived EBs (Supplementary Fig 5a, 5b). However, we did not see significant changes in the expression of differentiation markers such as *Gata4*, *Gata6*, *T-bra* and *Nestin* upon addition of Wnt3a. (Supplementary Fig 5c, 5d, 5e, 5f). Thus, while certain aspects of pluripotency were rescued upon exogenous addition of Wnt3a, not all features were rescued, indicative of the involvement of additional molecules and pathways that may be altered in the absence of CLCa.

[Figure removed by editorial staff per authors' request]

7. For a better understanding of the different pathways involved in early mESCs development, a schematic demonstration would be helpful for the reader who wants to see how the found effects are embedded in other signaling cascades.

Response: We have included a summary figure (Fig 6). To understand the various pathways involved in mESCs, we would like to refer the reviewer to some excellent reviews covering this topic. We have now included some of these references in our manuscript (Boyer et al.,2006; Tiwari et al., 2021; Varzideh et al., 2023).

Reviewer #2 (Comments to the Authors (Required):

1. An overarching critique of this work is that it relies heavily on the use of clonal cell lines, both for the mESC and the L-cell studies. Clonal variations unrelated to the specific alteration/mutation (*Clf* knockout or *Wnt3a* overexpression) may account for some, many or even all observed phenotypes. This is a common caveat in such studies and it is difficult to circumnavigate without analyzing multiple clones per genetic alteration. One possible way to address this critique is to use inducible systems. In the case of the *Clta/b* genes, an inducible shRNA knockdown system could be employed. For *Wnt3a*, an inducible overexpression system could be used. Such systems exist and should be incorporated into these studies.

Response: We have validated the phenotype observed in all CLC KO lines in 2 independent clones for each genotype (for mESCs as well as L-cells). Similar phenotypes were observed in the second clone generated for each cell type. We observed similar downregulated expression of Wnt target genes and reduction in β -catenin transcriptional activity (measured by Luciferase activity) (Supplementary Fig 4a,b,c) in the other clone of *Clta*^{-/-} and *Clta_b*^{-/-} mESCs as well. We also observed the appearance of actin patches in the second clone of *Clta*^{-/-} and *Clta_b*^{-/-} mESCs (Supplementary Fig 9) and L-cells (data not included in the manuscript, but shown below).

[Figure removed by editorial staff per authors' request]

2. Several experiments that examine Wnt3a trafficking would benefit from the use of a Porcn inhibitor as a control for Wnt secretion. Porcn inhibition (nearly) completely blocks Wnt secretion and observed effects may be more pronounced than what is observed in Clt Kos. This would be useful in assessing the extent of Wnt secretion blockage in the Clt KO lines. For example, in Figure 5a, it would be interesting to observe whether Porcn inhibition can reverse the actin organization of L-Wnt3a cells to that of L-cells. Wash-out of Porcn inhibitor could be employed as a means to do "pulse-chase" experiments.

Response: We thank the reviewer for this comment. To look at the effect of Porcupine inhibition on Wnt3a secretion, we treated Wnt3a L-cells with the Porcn inhibitor (ETC-159, 100nm) for 24h. This resulted in a dramatic reduction in secretion of Wnt3a compared to Wnt3a L-cells treated with DMSO control (Supplementary fig 8c). However, at 24h ETC-159 treatment was not sufficient to induce changes in the actin organization. We observed formation of actin patch-like structures in *Clta*^{-/-} Wnt3a L-cells and lesser patch formation in *Clta_b*^{-/-} Wnt3a L-cells after 72h treatment with the inhibitor (Supp. Fig. 13a). We treated WT and CLC KO Wnt3a L

cells with the inhibitor for 7 days to induce actin patch formation followed by washout for 72h and assessed the presence of actin patches. Washout for 72h after 7days treatment with the inhibitor resulted in the disappearance/reduction in the actin patches in *Clta*^{-/-} and *Clta_b*^{-/-} Wnt3a L-cells, indicating that there was indeed a direct effect of reduced Wnt3a secretion on actin organization. (Supplementary Fig 13b)

[Figure removed by editorial staff per authors' request]

[Figure removed by editorial staff per authors' request]

3. The loss of CLCs may also impact secretion and recycling of Wnt receptors, such as the Frizzleds and Lrp5/6. The fact that CLC knockout lines can still respond to exogenous Wnt3a treatment (Figure 1f) indicates that receptors are present and capable of transducing Wnt signals. Nonetheless, this should be examined.

Response: We thank the reviewer for this comment. We checked the total protein levels of Lrp6 in WT and CLC KO mESCs and it was found to be reduced in *Clt1*^{-/-} mESCs as compared to WTmESCs (Supplementary Fig 6c,d). However, we were not able to study the recycling of Lrp6 in these cells.

[Figure removed by editorial staff per authors' request]

4. Multiple experiments indicate that *Clta* plays a bigger role in Wnt trafficking than *Cltb* (e.g. Figures 1d, 2e,f, 3a-d), an observation that the authors point out but fail to pursue further. It would be interesting to test whether overexpression of *Clta* only can rescue the phenotypes observed in *Clta_b* double KO lines. The Discussion provides very little commentary on the distinct roles of these 2 genes in Wnt signaling and development.

Response: We thank the reviewer for the comment. To address the comment, we performed TIRF microscopy for the presence of Wnt3a at the PM after overexpressing CLCa and the control vector in the CLC KO Wnt3a L-cells. Overexpression of CLCa resulted in increased expression of Wnt3a at the PM compared to the cells transfected with control vector (Supplementary Fig 8d, e).

[Figures removed by editorial staff per authors' request]

5. In EBs, expression of pluripotency markers is elevated in *Clta* knockout cells, but not in *Cltb* or *Clta_b*. How do the authors explain that the increased levels observed in the single *Clta* KO line is not observed in the *Clta_b* double KO line? The same question applies to the elevated levels of differentiation markers (T, GATA4, AFP) in *Cltb* KO EBs: how is that this effect is not observed in the double KOs?

Response: Both light chains share 60% sequence similarity and have characteristic tissue specific expression. CLCa is exclusively expressed in spleen (Wu et al., 2016), in addition to being expressed widely in other tissues and organs. CLCb levels were upregulated in migratory trophoblast cells during invasion (Majeed et al., 2014) and non-small cell lung carcinoma (NSCLC) patients or cell lines (Chen et al., 2017)

In the single light chain knock out mESCs, the presence of the other light chain may compensate and skew the levels of either the pluripotency or differentiation genes depending on the signalling pathways it may regulate. In *Clta_b* double KO lines, the absence of both the CLCs, may further alter the preference towards a certain lineage. However, to comment on the involvement of a specific CLC in lineage commitment, a detailed analysis of the expression levels of CLCs in different tissues at various stages of development needs to be performed, which is clearly beyond the scope of this study. Since the CLCs have tissue specific expression and function perhaps the differences will appear upon performing directed differentiation studies of individual knockout mESCs in a temporal manner. This is however not within the scope of this study and will form the basis of future studies.

6. Supp Figure 1: Authors claim that there are no significant changes in expression of pluripotency markers between WT and *Clt* knockout lines, however, Oct4 levels appear to 2-3-fold higher, and in one setting *Klf4* expression is elevated 8-fold. Subtle changes such as these may in fact be meaningful. As shown it is difficult to appreciate the statistical significance (p-values?)

Response: We do observe upregulated levels of pluripotency markers in *Clta*^{-/-} mESCs. However due to variation in data values, this is not statistically significant. We have now mentioned the actual p values for the data included in this experiment.

[Figure removed by editorial staff per authors' request]

Minor:

1. Figure 1C is confusing. The color-coding of the "Group" on the top of the heatmap does not align with the indicated genotypes at the bottom of the heatmap.

Response: We apologize for this. We have now fixed this.

2. Line 133 and Figure 1D: Dkk1 expression is not reduced in *Clta*.

Response: We apologize for the error. Dkk1 expression is indeed reduced in *Clta*^{-/-} and *Clta_b*^{-/-} mESCs.

3. Line 181-182: what are the time points (6h, 12h, 24h, 48h) relative to? Was media replaced on cells at timepoint 0 and then harvested and quantified for Wnt3a levels at these timepoints? This is not explained well enough in the text or the figure legend.

Response: We have now included a schematic for how the experiment was performed.

[Figure removed by editorial staff per authors' request]

4. Although Wnt is in a complex with WLS, WLS is not a Wnt receptor. WLS is a Wnt binding factor required for Wnt secretion. It could also be referred to as a chaperone.

Response: We thank the reviewer for this comment. We now refer to WLS as Wnt ligand secretion mediator or carrier protein and have changed this in the manuscript as well.

5. Supp Figure 2: The GO terms are difficult to read. Please increase font sizes.

Response We have increased the size of the figure for the font to be legible.

6. Line 229: this reference needs to be fixed: (Xia and Kanchanawong, Cell Reports, 2019).

Response: We apologize for the error. This has been fixed.

Reviewer #3 (Comments to the Authors (Required)):

Significant Concerns:

1. The involvement of clathrin in regulating Wnt/ β -catenin signaling is not novel, as previously reported in the following reference:
o Clathrin regulates Wnt/beta-catenin signaling by affecting Golgi to plasma membrane transport of transmembrane proteins. Munthe E, Raiborg C, Stenmark H, Wenzel EM. J Cell Sci. 2020 Jul 9;133(13). doi: 10.1242/jcs.244467.

Response: The involvement of clathrin in regulating Wnt/ β -catenin signalling has always been controversial. At the very outset, we would like to clarify that previous reports involving clathrin and Wnt signalling all refer to the role of the clathrin heavy chain and do not describe the role of the light chain. It was previously shown that the clathrin heavy chain is involved in regulating Wnt/ β -catenin signaling by controlling exocytosis of transmembrane proteins Lrp5 and Lrp6 and members of cadherin

family from the Golgi (Munthe et al., 2020). Rim et al., 2020 have reported that knockdown of *Cltc* or knocking out components of the adaptor complex AP2 α in mESCs did not affect accumulation and nuclear translocation of β -catenin and Axin2 transcription upon treatment with exogenous Wnt3a. However, this study did not comment on the trafficking of components involved in the activation of the Wnt signaling pathway upon altering CME or the role of CLCs as opposed to *Cltc*. Loss of CLC does not affect gross CME. Our study emphasizes the involvement of CLCa in the trafficking of Wnt and WLS from the Golgi to the PM which is not well studied in the context of Wnt3a secretion along with the dependence on the proper organization of the actin cytoskeleton.

2. The manuscript does not discuss the status of Wnt5a/b in CLC KO mESCs, which are also regulated by WIs and involved in actin cytoskeletal organization.

Response: We thank the reviewer for this comment. We examined the total protein levels of Wnt5a in WT and CLC KO mESCs. It was found to be reduced in all CLC KO mESCs, but significantly reduced in *Cltc*^{-/-} mESCs. However treatment of *Cltc*^{-/-} and *Cltc_b*^{-/-} mESCs and L-cells with Wnt5a recombinant protein did not rescue the actin patch phenotype. (Supplementary Fig 14a, b, c, & d)

3. The importance of CLCa-regulated Wnt3a signaling and actin cytoskeletal organization in pluripotency, differentiation, and fate determination of mESCs is not well elucidated. Additionally, while CLCb appears to regulate endoderm lineage, its role in actin cytoskeletal organization is unclear.

Response: We thank the reviewer for this comment. In our study we have shown the involvement of CLCa in the secretion of Wnt3a in a WLS-dependent manner, and further the output of Wnt signalling activity. We further showed the requirement of CLCa in actin organization in a Hip1R-dependent manner, where total levels of Hip1R are maintained by Wnt signalling activity. This indicates a three-way connection between CLCa, Wnt signalling and actin organization. Wnt signaling plays a biphasic role during development. At early stages of development, in particular gastrulation, Wnt is required for mesoderm induction. However, if it is activated after gastrulation, it suppresses cardiac differentiation (Ueno et al., 2007). Similarly, actin plays a dynamic role through development. Our results hint at novel

modes of regulation of these pathways through the action of CLCa, regulating the secretion of Wnt3a in development. We observed that the expression of pluripotency markers Oct4 and Sox2 were reduced in *Clta*^{-/-} EBs upon treatment with Wnt3a (Supp Fig 5a, 5b). However, no significant alteration in the levels of other markers were observed in the knockout lines. *Cltb*^{-/-} mESCs showed a preference towards the endodermal lineage but did not show any change in actin organization. Despite sharing a common consensus sequence at the N-terminus in both the CLCs, they seem to differentially regulate actin organization. One may speculate that actin-dependent endocytic events in mESCs may prefer CLCa not CLCb. In support of this, the yeast clathrin light chain closely resembles CLCa rather than CLCb. Interaction with cortical actin through CLCs is essential for endocytosis in yeast. We believe that the differential role of mammalian CLCs towards actin may be due to functional differences in the sequences outside of common Hip1R binding region, which may modulate Hip1R-actin interactions.

4. The conclusion that Latrunculin-A suppresses Wnt signaling requires additional supporting data. Specifically, does Latrunculin affect Wls or Wnt3a expression in mESCs? Is it acting downstream to GSK3 β ?

Response: In this study we have shown that Latrunculin A treatment reduces the beta-catenin transcriptional activity using a TOP-FLASH reporter based on luciferase activity. To strengthen the role of the actin cytoskeleton in Wnt signalling, we treated WT and CLC KO Wnt3a L-cells with Lat A (0.1 μ m) for 6h and checked for secretion of Wnt3a through ELISA. We observed reduced secretion of Wnt3a upon Lat A treatment compared to Wnt3a L-cells treated with DMSO. (Fig 5g)

We have also shown loss of actin patches upon addition of CHIR99021 to *Clta*^{-/-} and *Clta_b*^{-/-} mESCs. We could also rescue actin organization in *Clta*^{-/-} L-cells upon treatment with CHIR99021 or recombinant Wnt3a but not in *Clta_b*^{-/-} L-cells, indicating that the cell type may further influence the effect of loss of clathrin light chains. Based on these, we speculate that affecting actin organization does indeed impact Wnt secretion and downstream signalling.

We were unable to check the levels of WLS or Wnt3a upon Lat A treatment in WT and CLC KO mESCs due to unavailability of a proper antibody against WLS and very low levels of Wnt3a in mESCs.

5. The preference of *Cltb*^{-/-} KO mESCs for endodermal lineage is noted, but it is unclear whether this is related to Wnt signaling or actin cytoskeletal organization or both.

Response: Both qRT-PCR and immunostaining data showed upregulated expression of alpha-fetoprotein (AFP) in *Cltb*^{-/-} mESCs derived EBs indicating their preference towards the endodermal lineage. Additionally, *Cltb*^{-/-} EBs also showed upregulated expression of cardiac genes resulting in increased number of beating EBs. Wnt signaling plays a biphasic role during cardiac differentiation. At early stages of development, in particular gastrulation, Wnt is required for mesoderm induction. However, if it is activated after gastrulation, it suppresses cardiac differentiation (Ueno et al., 2007). KEGG pathway enrichment analysis showed enrichment of the Wnt signaling pathway in both *Clta*^{-/-} and *Cltb*^{-/-} mESCs. This suggests a requirement of specific CLCs in regulating early developmental decisions to specify cell fate. We further speculate that this may arise due to differential trafficking of specific receptors. To address this, we treated EBs with Wnt3a and examined the expression of pluripotency and differentiation markers. We observed that the expression of the pluripotency markers Oct4 and Sox2 were reduced in *Clta*^{-/-} EBs upon treatment with Wnt3a (Supplementary Fig 5a,5b). However, no significant alteration in the levels of other markers were observed in the knockout lines. Modulation of the actin cytoskeleton with pharmacological inhibitors over the duration of EB generation was toxic and we were unable to perform these experiments to dissect the role of actin in this process. We believe that due to the dynamic requirements of both actin and Wnt at various stages of development, it is not possible to precisely comment on which of factors these skew the differentiation towards a specific lineage. To determine this, detailed spatial and temporal analysis through development would need to be performed, which is beyond the scope of this manuscript.

6. It would be helpful to clarify whether Wnt5a suppresses the actin patches found in CLCa KO mESCs.

Response: We thank the reviewer for suggesting this experiment. In both *Clta*^{-/-} and *Clta_b*^{-/-} mESCs and L cells, treatment with Wnt5a recombinant protein did not result in the suppression of actin patches. (Supplementary 14a & d). This indicates that while Wnt5a follows a similar secretory route to Wnt3a, the phenotypes observed upon loss of CLCa appear to be specific to Wnt3a.

[Figures removed by editorial staff per authors' request]

7. The role of HIP1R in actin cytoskeletal organization-whether influenced by its levels alone or also by its activity through phosphorylation or other modifications- requires further clarification.

Response: In mammalian cells, Hip1 and Hip1R undergo tyrosine phosphorylation by receptor tyrosine kinases (RTKs) such as Epidermal growth factor receptor (EGFR) and platelet-derived growth factor β receptor (PDGF β R) and are involved in cell survival and transformation (Ames et al., 2013). However, the role of phosphorylated Hip1R in actin organization in metazoans has not been described. Metazoan cells with reduced expression of Hip1R display altered actin polymerization at clathrin-coated pits (Engqvist-Goldstein et al., 2004; Le Clainche et al., 2007). Vertebrate Hip1R has also been proposed to be a negative regulator of actin polymerization at sites of clathrin assembly because of its ability to bind to the actin regulator, Cortactin (Le Clainche et al., 2007).

However, in *Dictyostelium* Hip1r lacks the cortactin-binding domain, and interaction between *Dictyostelium* epsin and Hip1r plays a role in the regulation of dynamic actin during clathrin-mediated endocytosis. Epsin modulates actin dynamics at clathrin-coated pits by facilitating the recruitment and phosphorylation of Hip1r (Brady et al., 2010). However, this regulation involving phosphorylation of Hip1R has not been described in metazoans and the majority of the actin regulation is believed to involve competition with cortactin.

8. The effect of CLCa on LRP6 expression in mESCs should be explored-

Response: We have now examined the levels of LRP6 and find that the total protein levels of Lrp6 were found to be reduced significantly in *Clta*^{-/-} mESCs. (Supplementary Fig 6c, d).

9. Does CLCb bind to HIP1R in mESCs?

Response: Both CLCa and CLCb have a 22 amino acid conserved sequence present at the N-terminus through which they bind to Hip1 and Hip1R. We have now performed co-immunoprecipitation experiments in WT mESCs, and find that both CLCa and CLCb bind to Hip1R. (Supplementary Fig 10).

[Figure removed by editorial staff per authors' request]

Minor Comments:

1. Several references cited in the text are missing from the reference list (e.g., Ferreira et al., 2012; Price et al., 2013; Tsygankova et al., 2019; Xia and Kanchanawong, Cell Reports, 2019, among others).

Response: We apologize for this error. We have now rectified this in the manuscript.

2. In Figure 6, Porcupine is mentioned, but the illustration does not depict or identify PORCN.

Response: We have corrected this now.

References:

1. Ames, H.M., Wang, A.A., Coughran, A., Evaul, K., Huang, S., Graves, C.W., Soyombo, A.A. and Ross, T.S., 2013. Huntingtin-interacting protein 1 phosphorylation by receptor tyrosine kinases. *Molecular and Cellular Biology*, 33(18), pp.3580-3593.
2. Boyer, L.A., Lee, T.I., Cole, M.F., Johnstone, S.E., Levine, S.S., Zucker, J.P., Guenther, M.G., Kumar, R.M., Murray, H.L., Jenner, R.G. and Gifford, D.K., 2005. Core transcriptional regulatory circuitry in human embryonic stem cells. *Cell*, 122(6), pp.947-956.

3. Brady, R.J., Damer, C.K., Heuser, J.E. and O'Halloran, T.J., 2010. Regulation of Hip1r by epsin controls the temporal and spatial coupling of actin filaments to clathrin-coated pits. *Journal of Cell Science*, 123(21), pp.3652-3661.
4. Engqvist-Goldstein, A.E., Zhang, C.X., Carreno, S., Barroso, C., Heuser, J.E. and Drubin, D.G., 2004. RNAi-mediated Hip1R silencing results in stable association between the endocytic machinery and the actin assembly machinery. *Molecular Biology of the Cell*, 15(4), pp.1666-1679.
5. Le Clainche, C., Pauly, B.S., Zhang, C.X., Engqvist-Goldstein, Å.E., Cunningham, K. and Drubin, D.G., 2007. A Hip1R–cortactin complex negatively regulates actin assembly associated with endocytosis. *The EMBO Journal*, 26(5), pp.1199-1210.
6. Tiwari, M., Bhattacharyya, S. and Subramanyam, D., 2021. Signaling pathways influencing stem cell self-renewal and differentiation. *Stem Cells and Aging* (pp. 69-87). Academic Press.
7. Varzideh, F., Gambardella, J., Kansakar, U., Jankauskas, S.S. and Santulli, G., 2023. Molecular mechanisms underlying pluripotency and self-renewal of embryonic stem cells. *International Journal of Molecular Sciences*, 24(9), p.8386.

March 20, 2025

RE: Life Science Alliance Manuscript #LSA-2024-02962-TR

Dr. Deepa Subramanyam
National Centre for Cell Science
SP Pune University
Ganeshkhind Road
Pune, Maharashtra 411007
India

Dear Dr. Subramanyam,

Thank you for submitting your revised manuscript entitled "CLCa mediates a novel cross-talk between Wnt secretion and actin organization". We would be happy to publish your paper in Life Science Alliance pending final revisions necessary to meet our formatting guidelines.

- please be sure that the authorship listing and order is correct
- please upload all figure files as individual ones, including the supplementary figure files; all figure legends should only appear in the main manuscript file
- please add the X and Bluesky handles of your host institute/organization as well as your own or/and one of the authors in our system
- please add your main, supplementary figure, and table legends to the main manuscript text after the references section
- please use the [10 author names, et al.] format in your references (i.e., limit the author names to the first 10)
- please upload your Tables in editable .doc or excel format
- we encourage you to revise the figure legends for figure S1 such that the figure panels are introduced in alphabetical order
- please remove label of panel A from figure S10 since it has only one panel
- you may want to upload Figure 6 as a Graphical Abstract rather than as a figure, but this is up to you

LSA now encourages authors to provide a 30-60 second video where the study is briefly explained. We will use these videos on social media to promote the published paper and the presenting author (for examples, see <https://docs.google.com/document/d/1-UWCfbE4pGcDdcgzcmiuJl2XMBJnxKYeqRvLLrLS08s/edit?usp=sharing>). Corresponding or first-authors are welcome to submit the video. Please submit only one video per manuscript. The video can be emailed to contact@life-science-alliance.org

A. FINAL FILES:

B. MANUSCRIPT ORGANIZATION AND FORMATTING:

Sincerely,

Reviewer #2 (Comments to the Authors (Required)):

I have re-read the manuscript and carefully reviewed the responses to comments/critiques from all reviewers. The overall conclusions have not substantially changed in the revision and I feel that the authors adequately addressed all concerns. I have no further critiques and am of the opinion that the manuscript is suitable for publication in LSA.

Reviewer #3 (Comments to the Authors (Required)):

The authors have thoroughly addressed all my concerns, and the revised manuscript is significantly improved compared to the original submission. I have no further concerns regarding this study.

April 17, 2025

RE: Life Science Alliance Manuscript #LSA-2024-02962-TRR

Dr. Deepa Subramanyam
National Centre for Cell Science
SP Pune University
Ganeshkhind Road
Pune, Maharashtra 411007
India

Dear Dr. Subramanyam,

Thank you for submitting your Research Article entitled "CLCa mediates a novel cross-talk between Wnt secretion and actin organization". It is a pleasure to let you know that your manuscript is now accepted for publication in Life Science Alliance. Congratulations on this interesting work.

DISTRIBUTION OF MATERIALS:

Again, congratulations on a very nice paper. I hope you found the review process to be constructive and are pleased with how the manuscript was handled editorially. We look forward to future exciting submissions from your lab.

Sincerely,
